# REAL3D-PORTRAIT: ONE-SHOT REALISTIC 3D TALKING PORTRAIT SYNTHESIS

**Zhenhui Ye** *†♠♡   **Tianyun Zhong** *†♠♡   **Yi Ren** ♡   **Jiaqi Yang** †♡   **Weichuang Li** ◇
**Jiawei Huang** †♠♡   **Ziyue Jiang** †♠♡   **Jinzheng He** †♠♡   **Rongjie Huang** ♠   **Jinglin Liu** ♡
**Chen Zhang** ♡   **Xiang Yin** ♡   **Zejun Ma** ♡   **Zhou Zhao** ‡♠
♠Zhejiang University & ♡ByteDance & ◇ HKUST(GZ)
{zhenhuiye,zhaozhou}@zju.edu.cn,
{ren.yi,yinxiang.stephen}@bytedance.com

## ABSTRACT

One-shot 3D talking portrait generation aims to reconstruct a 3D avatar from an unseen image, and then animate it with a reference video or audio to generate a talking portrait video. The existing methods fail to simultaneously achieve the goals of accurate 3D avatar reconstruction and stable talking face animation. Besides, while the existing works mainly focus on synthesizing the head part, it is also vital to generate natural torso and background segments to obtain a realistic talking portrait video. To address these limitations, we present Real3D-Potrait, a framework that (1) improves the one-shot 3D reconstruction power with a large image-to-plane model that distills 3D prior knowledge from a 3D face generative model; (2) facilitates accurate motion-conditioned animation with an efficient motion adapter; (3) synthesizes realistic video with natural torso movement and switchable background using a head-torso-background super-resolution model; and (4) supports one-shot audio-driven talking face generation with a generalizable audio-to-motion model. Extensive experiments show that Real3D-Portrait generalizes well to unseen identities and generates more realistic talking portrait videos compared to previous methods[1].

## 1   INTRODUCTION

Talking head generation aims to synthesize a talking portrait video given the driving condition (either a motion sequence (Wang et al., 2021) or a driving audio (Chung et al., 2017; Kim et al., 2019; Yi et al., 2020; Ye et al., 2022). It is a long-standing cross-modal task in computer graphics and computer vision with several real-world applications (Huang et al., 2022; 2023a) such as video conferencing Huang et al. (2023b) and visual chatbot Ye et al. (2023c). Previous 2D methods (Wang et al., 2021; Zhou et al., 2020; Lu et al., 2021) could produce photo-realistic videos thanks to the power of generative adversarial networks (GAN). However, due to the lack of explicit 3D modeling, these 2D methods are challenged with warping artifacts and unrealistic distortions at a significant head movement. In the past few years, neural radiance field (NeRF)-based 3D methods (Mildenhall et al., 2020; Guo et al., 2021; Hong et al., 2022b; Shen et al., 2022; Ye et al., 2023b) have been prevailing since they maintain realistic 3D geometry and preserve rich texture details even at a large head pose. However, among most of them, the model is over-fitted to a specific person, which requires expensive individual training for every unseen identity. It is promising to explore the task of one-shot 3D talking face generation, i.e., given an unseen person's reference image, we aim to lift it into a 3D avatar and animate it with the input condition to obtain a realistic 3D talking person video.

With recent advances in 3D generative models, it is possible to learn a hidden space of 3D triplane representation (EG3D, Chan et al. (2022)) that generalizes to various identities. While recent works (Li et al., 2023b; Li, 2023) have pioneered one-shot 3D talking face generation, they fail

---

*Equal contribution.

†Interns at ByteDance.

‡Corresponding author.

[1]Video samples and source code are available at https://real3dportrait.github.io

to achieve **accurate reconstruction** and **animation** simultaneously. To be specific, some works (such as OTAvatar, Ma et al. (2023)) first generate a 2D talking face video, then lift it into 3D via 3D GAN inversion (Yin et al., 2023), which enjoy rich 3D prior knowledge from the 3D GAN, yet are challenged with animation artifacts like temporal jitters and image distortion. Another line of works (Yu et al., 2023; Li et al., 2023a) learn a feed-forward network to predict the 3D representation from the image, then deform the 3D model with the input condition. However, the image-to-3D reconstruction process is not robust due to the lack of large-pose multi-view frames in video datasets, which are essential for learning 3D geometry. Due to the aforementioned challenges, the existing one-shot 3D methods cannot produce high-quality talking face videos.

Based on this observation, the first goal of this paper is to improve the *3D reconstruction and animation* power: (1) As for *reconstruction*, we propose to first pre-train a large Image-to-Plane (I2P) model by distilling 3D prior knowledge from a well-trained 3D face generative model. The I2P model is a feed-forward network that learns to reconstruct 3D representations of the input image with a single forward. We combine the advantage of ViT (Dosovitskiy et al., 2020) and VGGNet (Simonyan & Zisserman, 2014) to construct the network architecture and scale up the model to better store the knowledge of the image-to-3D mapping. (2) As for *animation*, we design an effective facial Motion Adapter (MA) to morph the predicted 3D representation given the input condition. Specifically, the motion adapter takes a fine-grained motion representation, projected normalized coordinate code (PNCC) (Zhu et al., 2016) as the input, then predicts a residual motion diff-plane that edits the reconstructed 3D representation via element-wise addition. Since PNCC is well disentangled from appearance and pose, we use a shallow SegFormer (Xie et al., 2021) to inject the input expression information into the canonical space efficiently. In a word, we pre-train a large-scale image-to-plane backbone for generalized and high-quality 3D reconstruction, then utilize a lightweight motion adapter to achieve efficient face animation.

The second goal is to improve the naturalness of the synthesized torso and background segments. Existing methods either only model the head part (Hong et al., 2022b) or model the head and torso as a whole (Li et al., 2023b), which overlook the necessity of a natural torso and background to obtain a realistic talking portrait video. To handle this limitation, we propose to individually model the head, torso, and background segments and compose them into the final image during the rendering process. Specifically, we design a Head-Torso-Background Super-Resolution (HTB-SR) model, which consists of a super-resolution branch to upsample the low-resolution volume-rendered head images, a warping-based torso branch for modeling the individual torso movement, as well as a background branch to achieve switchable background rendering. With these designs, we could render realistic and high-fidelity 3D talking portrait video given the motion condition. To further support audio-driven applications, we design a generic variational audio-to-motion (A2M) model to transform the audio signal into the motion representation PNCC. Our audio-to-motion model generalizes well to unseen identities without adaptation and supports explicit eye blink and mouth amplitude control.

To summarize, in this paper, we propose **Real3D-Portrait**, a one-shot and **Real**istic **3D** talking **Portrait** generation method that: (1) improves the *3D reconstruction and animation* power with I2P model and motion adapter; (2) achieves natural torso movement and switchable background rendering with HTB-SR model; and (3) proposes a generic A2M model, hence becomes the first one-shot 3D face system that both supports audio and video-driven scenarios. Experiments show that our method outperforms existing one-shot talking face systems and achieves comparable performance to state-of-the-art person-specific methods. Ablation studies prove the effectiveness of each component.

## 2 RELATED WORK

Our work focuses on the task of one-shot 3D talking face generation, it mainly relates to two aspects of *reconstruction* and *animation*, i.e., (1) How to reconstruct an accurate 3D face representation of the input image; (2) How to morph the 3D representation and render the talking face that corresponds to the driving condition (motion or audio). We discuss them respectively in the following sections.

**3D Face Representation**    Introducing 3D face representation into the talking face generation is a fundamental technique to improve the naturalness of the synthesized video. The earliest adopted 3D representation is the 3D Morphable Model (3DMM) (Blanz & Vetter, 1999), which provides a strong geometry prior to the face rendering process. However, the accuracy of 3DMM is known

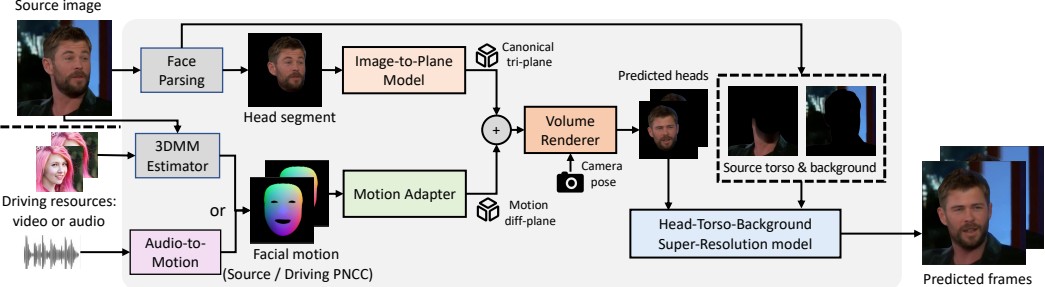

Figure 1: The inference pipeline of Real3D-Portrait. With one source image as input and a video/audio as driving condition, it synthesizes 3D talking avatars with a realistic torso and background.

to be unsatisfactory due to two limitations: (1) the reconstructed face mesh is of low fidelity and lacks details such as wrinkles; (2) only the face region is modeled by the parametric model, and it fails to represent other regions, such as hair, hat, and eyeglass. Then, a 3D head representation based on Neural Radiance Fields (NeRFs) (Mildenhall et al., 2020) emerged. Early NeRF-based 3D representations are typically extracted in an inefficient per-person-per-training manner, which takes tens of hours to fit each identity. Recently, the invention of tri-plane representation (Chan et al., 2022) and its usage in 3D face GAN paves the way for high-quality and efficient NeRF-based 3D face reconstruction. Some works (Sun et al., 2023) utilize GAN inversion (Roich et al., 2022) to obtain a tri-plane from the pretrained 3D Face GAN, which suffers from slow inference and temporal jittering. In contrast, other predictor-based works (Trevithick et al., 2023) explore learning an image-to-plane mapping that directly maps the input image to the tri-plane representation, which is more efficient and stable during inference. Our large image-to-plane model follows the predictor-based paradigm.

**2D/3D Face Animation** The earliest 2D-based face animation methods like (Prajwal et al., 2020; Hong et al., 2022a) directly adopt GAN to generate the result, which results in training instability and bad visual quality. Later, the warping-based methods (Siarohin et al., 2019; Wang et al., 2021; Zhao & Zhang, 2022; Pang et al., 2023; Hong & Xu, 2023) aim to warp the pixels of the source image with a dense warping field given the 3D-aware key points extracted from the driving video. It achieves high image fidelity, yet due to the absence of a strict 3D constraint, it is challenged when driven by a large head pose and suffers from warping artifacts and distortions. To handle the artifacts caused by 2D modeling, some works resort to 3D-based methods.

The earliest 3D talking face methods are primarily based on the 3DMM, which typically first performs 3D reconstruction of the input image (Deng et al., 2019c; Daněček et al., 2022), then incorporates the 3DMM prior into the face rendering process (Wu et al., 2021). However, these methods fail to generate photo-realistic results due to the information loss caused by 3DMM. Recently, NeRF-based talking face generation has prevailed since it combines the advantages of high image fidelity and strict geometry constraints. However, most of the successful ones are identity-overfitted (Guo et al., 2021; Tang et al., 2022; Ye et al., 2023a), which requires tens of hours of individual training for every unseen identity. Most recently, some works explore one-shot NeRF-based talking face generation with the tri-plane representation, which can be categorized into two classes. The first class (Ma et al., 2023; Li, 2023; Trevithick et al., 2023) adopt a *2D animation and 3D lifting* pipeline, which utilizes a pre-trained 2D talking face system to obtain a 2D talking face video, then lift it into 3D via iterative 3D GAN inversion (Yin et al., 2023). This line of work could enjoy the robust 3D prior knowledge of a pre-trained 3D GAN but is also challenged by the unstable GAN inversion and degraded performance at large head poses. The second class adopts a *3D reconstruction and 3D animation* pipeline (Li et al., 2023a;b), which learns an image encoder to predict the 3D representation, then morphs the reconstructed 3D model given the condition. However, since video datasets typically lack large-view frames, the generalizability of 3D reconstruction is unsatisfactory. Due to space limitation, we discuss the relationship between our approach and previous methods in Appendix A.

## 3   REAL3D-PORTRAIT

Real3D-Portrait aims to achieve realistic one-shot video/audio-driven 3D talking face generation. As shown in Fig. 1, the overall inference pipeline is composed of a large image-to-plane (I2P) model

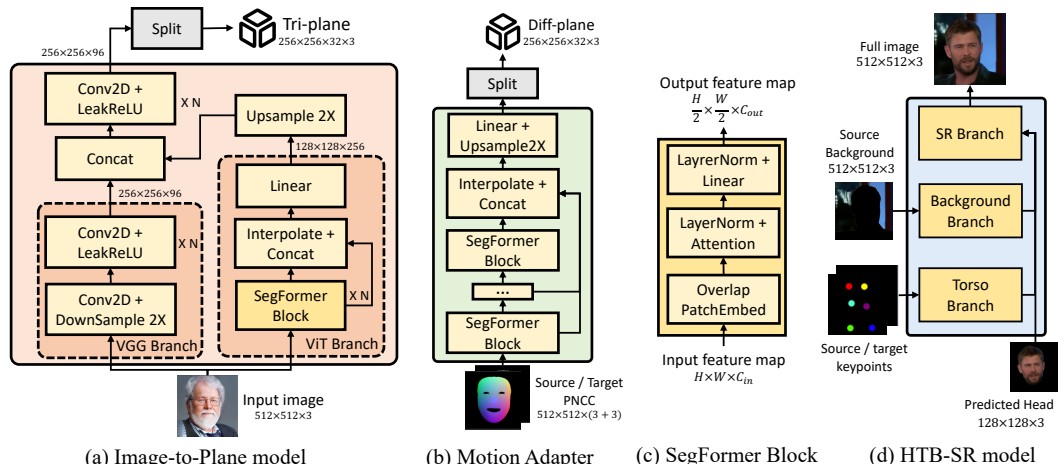

Figure 2: The network structure of I2P model, motion adapter, and HTB-SR model.

(Sec. 3.1) to reconstruct a 3D head representation and a motion adapter (Sec. 3.2) to morph the 3D head given the facial motion. Then, we could render the head image at an arbitrary camera (head) pose with the volume renderer. Afterward, we propose a head-torso-background super-resolution (HTB-SR) model (Sec. 3.3) to synthesize the final image at $512\times512$ resolution with individually modeled torso and background. To support audio-driven applications, we also design a generic audio-to-motion (A2M) model (Sec 3.4) to transform the raw audio into the corresponding facial motion. The training process of the four models is sequential. We describe the designs and training process in detail.

## 3.1 IMAGE-TO-PLANE MODEL FOR 3D FACE RECONSTRUCTION

In the first stage, we need to reconstruct a canonical 3D face representation $\mathbf{P}_{\text{cano}}$ of the target identity in the source image $\mathbf{I}_{\text{src}}$. Specifically, we learn a feed-forward network that directly transforms the input image into the tri-plane representation, namely the Image-to-Plane (I2P) model.

**Network Design** When designing the network structure of the I2P model, we notice two main challenges for the network: (1) it should map the input image to a canonical tri-plane, which requires a coordinate transform from pixel coordinate to world coordinate. (2) It should extract rich appearance features from the source image to guarantee the fidelity of the rendered image. To this end, as shown in Figure 2(a), we design a hybrid network consisting of a ViT and a VGG branches. The ViT branch comprises a stack of SegFormer blocks, which executes attention among patches and could efficiently handle the pixel-to-world canonicalization process. Since ViT cannot maintain high-frequency texture due to the patch embedding operation, as a complementary, we design a VGG branch, which is simply a stack of convolution layers to extract high-frequency appearance features. The two branches' output information is fused via concatenation and further processed with shallow convolution layers to produce the final tri-plane representation. Note that we remove all normalization in the VGG branch to keep the identity-specific appearance-related bias in all hidden layers.

**Pre-training Process** The talking face dataset typically lacks multi-view frames, which are necessary for the model to learn 3D prior knowledge. To improve the generalizability of 3D face reconstruction, inspired by the *multi-view reconstruction* task proposed by Trevithick et al. (2023), we first pre-train the I2P model on a multi-view image dataset synthesized by EG3D (Chan et al., 2022), a 3D GAN for generating human face. We illustrate the pre-training process in Appendix B.1.

## 3.2 MOTION ADAPTER FOR 3D FACE ANIMATION

With the pre-training process of the I2P model, we achieve to reconstruct an accurate 3D face representation from the source image. Then, we train a motion adapter to animate the predicted 3D face, given the input motion condition.

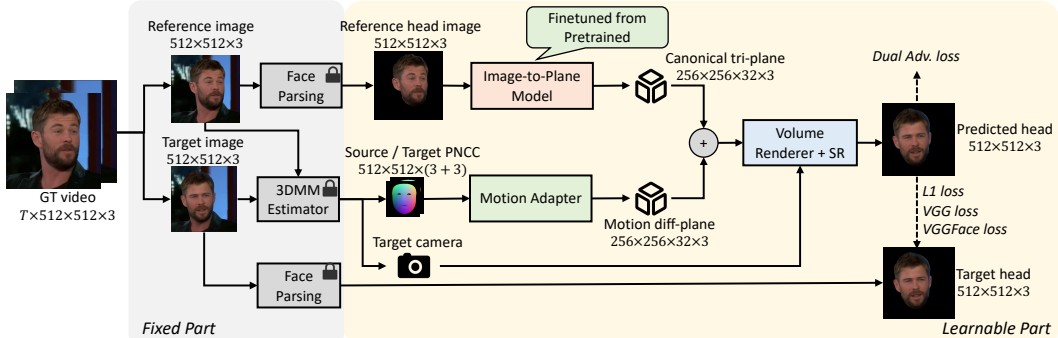

Figure 3: The process of training the motion adapter and fine-tuning the I2P model in Sec.3.2.

**Motion Representation**  We use the projected normalized coordinate code (PNCC) (Zhu et al., 2016; Kim et al., 2018; Li et al., 2023a) as the motion representation, which is a pose/appearance-agnostic feature map that possesses fine-grained facial expression information based on a 3DMM face. Specifically, given a pair of identity code $\mathbf{i}$ and expression code $\mathbf{e}$, we could obtain the PNCC by rasterizing the 3DMM face mesh at the canonical pose through Z-Buffer (Phong, 1998) algorithm with NCC (Zhu et al., 2016) as its colormap. We provide details of calculating PNCC in Appendix B.2.

Thanks to the identity-expression decomposition of 3DMM, we can utilize PNCC to achieve identity-agnostic motion-conditioned face animation. During training, we first fit the 3DMM parameters of the training video to obtain the ground truth PNCC. During inference, we could construct the driving PNCC by:

$$\mathbf{PNCC}_{\text{drv}} = \text{Z-Buffer}(3\text{DMM\_Mesh}(\mathbf{i}_{\text{src}}, \mathbf{e}_{\text{drv}}), \mathbf{NCC}), \tag{1}$$

where $\mathbf{i}_{\text{src}}$ is the identity coefficient of the source image, and $\mathbf{e}_{\text{drv}}$ is the expression coefficient that is either extracted from a driving video or predicted by an audio-to-motion model (Sec. 3.4).

**Predicting a Residual Motion Diff-plane**  Once the motion representation is decided, the second question is how to inject the motion condition into the 3D representation to control the facial expression. We do not choose the deformation field as previous works do since it typically results in bad quality of the predicted mesh (Li et al., 2023a). Instead, given a well-trained I2P model to produce a 3D representation that possesses accurate geometry/texture information, we propose to learn a light-weight Motion Adapter (MA) to predict a *residual motion diff-plane* $\mathbf{P}_{\text{diff}}$ that only edits the minimal geometry change of the canonical tri-plane $\mathbf{P}_{\text{cano}}$ based on the different motion condition. As for the network structure of the proposed MA, as shown in Fig. 2(b), we adopt a shallow SegFormer (Xie et al., 2021) to enjoy its high efficiency and strong ability to achieve cross-coordinate transform brought by the attention over feature map patches. To be specific, the process of animating the source image $\mathbf{I}_{\text{src}}$ given the input motion condition $\mathbf{PNCC}_{\text{drv}}$ and camera pose $\mathbf{cam}$ can be expressed as:

$$\mathbf{I_{drv}} = \text{SR}(\text{VR}(\mathbf{P}_{\text{cano}} + \mathbf{P}_{\text{diff}}, \mathbf{cam})), \ \ s.t., \ \mathbf{P}_{\text{cano}} = \text{I2P}(\mathbf{I}_{\text{src}}), \mathbf{P}_{\text{diff}} = \text{MA}(\mathbf{PNCC}_{\text{drv}}, \mathbf{PNCC}_{\text{src}}), \tag{2}$$

where VR and SR are a vanilla volume renderer and super-resolution module, whose structures are shown in Fig. 5; I2P and MA are the proposed image-to-plane model and motion adapter, respectively; $\mathbf{PNCC}_{\text{drv}}$ are the driving motion representation defined in Eq. 1 and $\mathbf{PNCC}_{\text{src}}$ is the PNCC extracted from the source image. Note that the input of MA is the concatenation of $\mathbf{PNCC}_{\text{src}}$ and $\mathbf{PNCC}_{\text{drv}}$. It is since the tri-plane predicted by the I2P model is of the source expression and the MA should be aware of both the source/driving expression to correctly map the 3D representation of the source expression into the target expression.

**Training Process**  We illustrate the training process at this stage in Fig. 3. Apart from learning the motion adapter from scratch, we also fine-tuned the I2P model and the VR/SR module from the pre-trained weights in Sec. 3.1. We train the model on a large-scale and high-fidelity talking face video dataset (Zhu et al., 2022). To construct the training data pair, we randomly select two frames from a video and define them as source image $\mathbf{I}_{\text{src}}$ and target image $\mathbf{I}_{\text{tgt}}$. Since camera pose is only highly correlated with the movement of the head part, we only consider the head region at this stage, so all images are preprocessed with a face parsing model to extract the head segment. The source

head image is fed into the I2P model to reconstruct a canonical tri-plane $\mathbf{P}_{\text{cano}}$, and the $\mathbf{PNCC}_{\text{tgt}}$ extracted from $\mathbf{I}_{\text{tgt}}$ is fed into the motion adapter MA to obtain the residual motion diff-plane $\mathbf{P}_{\text{diff}}$. Then we could obtain the predicted image $\mathbf{I}'_{\text{tgt}}$ following Eq. 2. The training loss is as follows:

$$\mathcal{L} = ||\mathbf{I}_{\text{tgt}} - \mathbf{I}'_{\text{tgt}}||_1^1 + \mathcal{L}_{\text{VGGs}}(\mathbf{I}_{\text{tgt}}, \mathbf{I}'_{\text{tgt}}) + \mathcal{L}_{\text{DualAdv}}(\mathbf{I}'_{\text{tgt}}) + \mathcal{L}_{\text{Lap}} \qquad (3)$$

where the first two terms are L1 loss and VGG19/VGGFace-based (Simonyan & Zisserman, 2014; Parkhi et al., 2015) perceptual loss; $\mathcal{L}_{\text{DualAdv}}$ is the dual adversarial loss proposed by (Chan et al., 2022) to improve the image fidelity and consistency; and $\mathcal{L}_{\text{Lap}}$ is our proposed Laplacian loss over the motion diff-planes of adjacent frames, which acts as a regularization term to eliminate temporal jittering. To be intuitive, given PNCCs from frame $\{t-1, t, t+1\}$, we expect the diff-plane $\mathbf{P}_{\text{diff}}$ of frame $t$ to be the average of that of $t-1$ and $t+1$:

$$\mathcal{L}_{\text{Lap}} = ||\mathbf{MA}(\mathbf{PNCC}_t) - 0.5 \times (\mathbf{MA}(\mathbf{PNCC}_{t-1}) + \mathbf{MA}(\mathbf{PNCC}_{t+1}))||_2^2 \qquad (4)$$

### 3.3 HEAD-TORSO-BACKGROUND SUPER-RESOLUTION MODEL

With the proposed I2P model and motion adapter, we could synthesize 3D talking heads given the source image and driving motion. The last step towards a realistic talking person video is synthesizing the torso and background segments. A naive solution is to model the torso/background along with the head using NeRF. However, applying the same rigid transformation to both the head and other regions results in unsatisfactory results (e.g., the torso and background will rotate along with the head movement). To generate a realistic torso and background, we propose a Head-Torso-Background Super-Resolution (HTB-SR) model to individually model the head, torso, and background segments and fuse them into a high-resolution composite image.

**Network Struture** As shown in Fig. 6 of Appendix B.3, the HTB-SR model consists of an SR branch, a Torso branch, and a Background branch. (1) The **SR branch** shares a similar structure with the vanilla SR module used in Sec. 3.2 so we could initialize its weights from the pre-trained model. (2) As for the **Torso branch**, since the movement of the torso is of small amplitude and often translational, we propose to model the torso part with a 2D warping-based renderer that is similar to (Wang et al., 2021), which is computationally efficient and proven robust in various scenes (Siarohin et al., 2019). As shown in Fig. 6(b), the torso segments are warped by dense flows conditioned on predefined key points to predict the torso feature map of the target image. Note that instead of learning implicit key points in an unsupervised manner as (Wang et al., 2021), we select several key points in the reconstructed 3DMM face vertex as the driving condition of the torso branch, which improves the temporal stability of the predicted torso. We provide details of the torso branch in Appendix B.3. (3) As for the **Background branch**, the biggest challenge is to fill the pixels that were occupied by the foreground (i.e., the person) in the source image. To this end, as shown in Fig. 6(c), we first adopt a K-nearest-neighbor (KNN)-based inpainting method [2] to preprocess the background segment of the source image. Then, we use shallow convolution layers to extract texture features from the inpainted background. More details about the background branch can be found in Appendix B.4. (4) As for **fusing the three feature maps** of head, torso, and background segments into a composite image, we found a direct channel-wise concatenation leads to hollow artifacts and blurry results in the boundary region (shown in Fig. 16). We suppose that the problem is caused by the unlimited information propagation among these three segments' feature maps and handle this problem with a **alpha-blending-style fusion** mechanism. To be specific, we first obtain the head mask $\mathbf{M}_{\text{head}}$ and torso mask $\mathbf{M}_{\text{torso}}$, then integrate the three segments with the awareness of occlusion:

$$\mathbf{F} = (\mathbf{F}_{\text{head}} \cdot \mathbf{M}_{\text{head}} + \mathbf{F}_{\text{torso}} \cdot (1 - \mathbf{M}_{\text{head}})) \cdot \mathbf{M}_{\text{person}} + \mathbf{F}_{bg} \cdot (1 - \mathbf{M}_{\text{person}}) \qquad (5)$$

where $\mathbf{F}$ denotes the extracted feature map, $\mathbf{M}_{\text{person}}$ is the person mask obtained by bitwise-or operation to $\mathbf{M}_{\text{head}}$ and $\mathbf{M}_{\text{torso}}$. Details of obtaining $\mathbf{M}_{\text{head}}$ and $\mathbf{M}_{\text{torso}}$ can be found in Appendix B.5.

**Training Process** As shown in Fig. 7, we load the pre-trained I2P model, motion adapter, and volume renderer from Sec. 3.2 and replace the SR module with our HTB-SR model. Only the HTB-SR model is updated at this stage, and all other parameters are frozen. The training objective is similar to Eq. 3. The difference is that the GT and predicted images are full images with head/torso/background parts instead of only the head segment.

---

[2]We can also use state-of-the-art neural network-based inpainting methods to obtain a more realistic background image. But since that is not the focus of this paper, we simply used the naive KNN-based method.

## 3.4 Generic Audio-to-Motion Model

To support audio-driven applications, we design a generic and controllable audio-to-motion (A2M) model to transform the audio into the PNCC motion representation. Inspired by Ye et al. (2023b), we adopt a flow-enhanced variational auto-encoder (VAE) to learn an accurate and expressive audio-to-motion mapping. HuBERT (Hsu et al., 2021) is chosen as the audio representation. As for the predicted motion representation, instead of directly predicting the PNCC, we choose to predict the 3DMM expression parameter, which utilizes the strong geometry prior of 3DMM and significantly improves the training stability and audio-lip accuracy. We choose BFM 2009 (Paysan et al., 2009) as the 3DMM model. Since all expression bases are orthogonal, given the same identity code, the reconstructed 3D face meshes in a video are uniquely determined by the expression code. Hence an L2 error on the expression code, $\mathcal{L}_{\text{ExpRecon}}$, is feasible to be the reconstruction term in training the VAE. To encourage the model to better reconstruct the facial landmark (instead of only the 3DMM parameters), we additionally introduce the L2 reconstruction error of 468 key points of the reconstructed 3DMM vertex, $\mathcal{L}_{\text{LdmRecon}}$, as an auxiliary supervision signal. The training loss of the generic audio-to-motion model is as follows:

$$\mathcal{L}_{\text{A2M}} = \mathcal{L}_{\text{KL}} + \mathcal{L}_{\text{ExpRecon}} + \mathcal{L}_{\text{LdmRecon}} + \mathcal{L}_{\text{ExpLap}} \tag{6}$$

where $\mathcal{L}_{\text{KL}}$ is the KL divergence of VAE; $L_{\text{expLap}}$ is the laplacian loss of the predicted expression code sequence to eliminate temporal jittering. To further improve the controllability, we add the eye blink and mouth amplitude as the auxiliary condition to the A2M model, which improves the expressiveness of the generated video. We provide detailed structure of A2M model in Appendix B.6

**Audio/Video-Driven Inference**  Once the four-stage training process is done, no further training is required for a new identity. During inference, as shown in Fig. 1, we first fit the 3DMM parameters of the source image to obtain the source identity code $\mathbf{i}_{\text{src}}$. As for audio-driven scenarios, we obtain the expression sequence $\mathbf{e}_{\text{drv}}$ corresponding to the input audio using the A2M model; as for the video-driven scenarios, we fit 3DMM on the reference video to obtain the $\mathbf{e}_{\text{drv}}$. Then, we could obtain the driving PNCC following Eq. 1 and render the final images following Eq. 2 and Eq. 5.

## 4 Experiment

### 4.1 Experimental Setup

**Implementation Details.** We provide detailed configuration and hyper-parameters in Appendix C, and will release the source code at `https://real3dportrait.github.io` in the future.

**Data Preparation.** To pre-train the I2P model, we adopt a 3D face generative model (Chan et al., 2022) to on-line generate multi-view image pairs during training. To train the motion adapter and HTB-SR model, we use a high-fidelity talking face video dataset, CelebV-HQ (Zhu et al., 2022), which is about 65 hours and contains 35,666 video clips with a resolution of $512 \times 512$ involving 15,653 identities. To train the A2M model, we use VoxCeleb2 (Chung et al., 2018), a low-fidelity but 2,000-hour-long large-scale lip-reading dataset to guarantee the generalizability of the audio-to-motion mapping. We preprocess the video frames with an off-the-shelf landmark extractor and face parser (Lugaresi et al., 2019), then fit 3DMM parameters based on the projected landmark error. We extract HuBERT features (Hsu et al., 2021) and pitch contours from the audio track.

**Compared Baselines.** We compare our Real3D-Potrait with several video/audio-driven baselines: 1) *Face-vid2vid* (Wang et al., 2021), a widely used warping-based video-driven talking face system; 2) *OT-Avatar* (Ma et al., 2023), a recent one-shot video-driven method that utilizes a pre-trained 3D GAN to obtain a 3D talking video; 3) *HiDe-NeRF* (Li et al., 2023a), a state-of-the-art one-shot 3D talking face system that utilizes deformation field for face animation; 4) *MakeItTalk* (Zhou et al., 2020) and 5) *PC-AVS* (Zhou et al., 2021), which are two one-shot audio-driven talking face method that achieve good audio-lip synchronization; 6) *RAD-NeRF* (Tang et al., 2022), a NeRF-based method that achieves high-realistic quality by over-fitting on the target person video. Note that it is unfair to compare RAD-NeRF against other one-shot methods, but we compare with it to show how far we are from the performance of the state-of-the-art person-specific method. We summarize the characteristics of all test baselines and our method in Table. 5.

Table 1: Same/Cross-identity reenactment results of video-driven methods. Best scores are in **bold**.

| Methods | Same-Identity Reenactment | | | | | | | | | Cross-Identity Reenactment | | | |
| --- | --- | --- | --- | --- | --- | --- | --- | --- | --- | --- | --- | --- | --- |
| | L1↓ | PSNR↑ | SSIM↑ | LPIPS↓ | FID↓ | CSIM↑ | AED↓ | APD↓ | AKD↓ | CSIM↑ | FID↓ | AED↓ | APD↓ |
| Face-vid2vid | 0.078 | 16.47 | 0.779 | 0.184 | 42.96 | 0.808 | 0.116 | 0.023 | 3.176 | 0.726 | 45.18 | 0.144 | 0.029 |
| OTAvatar | 0.106 | 13.17 | 0.672 | 0.232 | 65.37 | 0.568 | 0.170 | 0.040 | 5.891 | 0.544 | 64.28 | 0.195 | 0.046 |
| HiDe-NeRF | 0.084 | 15.92 | 0.752 | 0.189 | 50.04 | 0.753 | 0.129 | 0.021 | 3.531 | 0.699 | 53.28 | 0.161 | 0.025 |
| Ours (VD) | **0.067** | **18.95** | **0.801** | **0.171** | **37.50** | **0.821** | **0.111** | **0.018** | **2.829** | **0.758** | **42.37** | **0.138** | **0.022** |

## 4.2 QUANTITATIVE EVALUATION

Real3D-Portrait supports both video and audio as driving resources. In this section, we evaluate it and the baselines for video-driven reenactment and audio-driven talking face generation, respectively.

**Video-driven same/cross-identity reenactment.**   In the video-driven (VD) scenario, the driving motion condition and head pose are obtained from a reference video. Under the same-identity setting, we use the first frame of the reference video as the source image; otherwise, the source image is of a different identity. As for the same-identity setting, we evaluate PSNR, SSIM, cosine similarity of the identity embedding (CSIM) by Deng et al. (2019a), average expression distance (AED) and average pose distance (APD) based on (Deng et al., 2019b), average keypoint distance (AKD) based on (Bulat & Tzimiropoulos, 2017), as well as LPIPS, L1 and FID between the reenacted and ground truth frames. As for the cross-identity setting, since there is no ground truth, we evaluate the results based on the CSIM, AED, APD, and FID metrics. The results are shown in Table 1. Firstly, our video-driven Real3D-Portrait performs best in terms of L1, PSNR SSIM, LPIPS, and FID, hence achieving the best image quality. Secondly, our Real3D-Portrait gets the highest CSIM, which denotes that it preserves the identity of the source image. Finally, our method achieves the best AED, APD, and AKD, demonstrating that ours could accurately animate the 3D avatar given the input condition.

**Audio-driven talking face generation.**   In the audio-driven (AD) setting, the driving motion conditions are predicted from the input audio. Similar to the cross-identity reenactment, since there are no ground truth samples, we use CSIM to measure the identity preservation, FID to measure the image quality, and

Table 2: Results of audio-driven methods.

| Methods | CSIM↑ | FID↓ | AED↓ | Sync↑ |
| --- | --- | --- | --- | --- |
| MakeItTalk | 0.715 | 52.65 | 0.213 | 3.286 |
| PC-AVS | 0.327 | 82.02 | 0.162 | 6.483 |
| RAD-NeRF | **0.784** | **39.45** | 0.197 | 3.779 |
| Ours (AD) | 0.763 | 43.02 | **0.146** | **6.565** |

AED and Sync score (Prajwal et al., 2020) to measure the audio-lip synchronization. The results are shown in Table 2. When compared with MakeItTalk and PC-AVS, two one-shot 2D methods, our method shows significantly better identity similarity (CSIM), image quality (FID), and lip-synchronization (AED and Sync score). When compared with RAD-NeRF, a person-specific 3D method that over-fits an individual model on the tested identity's 3-minute-long video, apart from achieving better lip synchronization, our method remarkably shows comparable image quality and identity preserving thanks to the well-trained large I2P model and motion adapter. To summarize, the experiment demonstrates that our one-shot Real3D-Portrait outperforms other one-shot baselines and could perform closely to the SOTA person-specific over-fitting method.

## 4.3 QUALITATIVE EVALUATION

**Case Study**   To make a clear comparison among all tested methods, we provide demo videos at `https://real3dportrait.github.io`. We also provide more visualization results in Appendix D.3. Specifically, (1) we showcase the *overall qualitative comparison* of our Real3D-Portrait and other VD/AD baselines in Fig. 9 and Fig.10. We also provide examples to show that: (2) *how PNCC animates the 3D avatar* in Fig. 11; (3) we achieve *natural torso movement under large head poses* in Fig. 12; (4) we support *switchable background* in Fig. 13; (5) our *generic audio-to-motion model predicts synchronized lip motion* in Fig. 14.

**User study**   We conduct user studies to test the quality of generated samples. Specifically, we sample 10 audio clips from different languages and ten different identities for all methods to generate the videos and then involve 20 attendees for user studies. We adopt the Mean Opinion score (MOS) rating protocol for evaluation, which is scaled from 1 to 5. Following (Chen et al., 2020) The attendees are required to rate the videos from three aspects: (1) *identity preserving*; (2) *visual quality* (including image fidelity and temporal smoothness); (3) *lip synchronization*. Detailed user study settings are in Appendix D.2. We compute the average score for each method, and the results are shown in Table

Table 3: MOS score of different methods. The error bars are 95% confidence interval.

| Methods | Face-v2v | OTAvatar | HiDe-NeRF | **ours (VD)** | MakeItTalk | PC-AVS | RAD-NeRF | **ours (AD)** |
|---|---|---|---|---|---|---|---|---|
| ID. Preserving | 3.79±0.34 | 3.29±0.24 | 3.74±0.31 | **4.08±0.31** | 3.38±0.44 | 3.21±0.36 | **4.12±0.26** | 4.05±0.27 |
| Visual Quality | 3.73±0.25 | 3.28±0.29 | 3.45±0.29 | **4.16±0.23** | 3.34±0.34 | 3.40±0.37 | **4.25±0.24** | 4.14±0.29 |
| Lip Sync. | 3.97±0.20 | 3.80±0.28 | 3.54±0.32 | **4.13±0.29** | 2.96±0.37 | 4.04±0.31 | 3.18±0.52 | **4.08±0.25** |

3. We have the following observations: 1) Real3D-Portrait has better identity-preserving power and visual quality than previous one-shot methods and performs closely to person-specific methods (RAD-NeRF). 2) As for the lip-synchronization, Real3D-Portrait shows obvious superiority over the person-specific audio-driven method RAD-NeRF thanks to the powerful generic audio-to-motion model. Besides, among the video-driven methods that use GT motion as input, ours achieves the best lip synchronization, showing the effectiveness of our motion adapter to animate the avatar given the input motion accurately.

## 4.4 ABLATION STUDIES

**I2P and motion adapter**   We test four settings on the I2P and motion adapter: (1) *w/o pre-training*, which does not pre-train the I2P model on the multi-view image dataset in Sec.3.1; (2) *w/o fine-tuning*, which fix the pre-trained I2P model when training on the video dataset in Sec.3.2; (3) *small/large I2P model size*, which tries I2P backbone of difference scales of 40M and 200M parameters (note that the default setting is 87M parameters); (4) *w/o Lap loss*, which removes the laplacian loss in Eq.3.2. We show the result in Table 4. As shown in line 1 and Fig.

Table 4: Ablation studies.

| Methods | CSIM↑ | FID↓ | AED↓ | APD↓ |
|---|---|---|---|---|
| w/o pre-train | 0.487 | 65.32 | 0.181 | 0.031 |
| w/o finetune | 0.683 | 49.21 | 0.233 | 0.027 |
| w/ 40M params | 0.725 | 45.48 | 0.140 | 0.026 |
| w/ 200M params | 0.754 | 43.15 | 0.143 | 0.023 |
| w/o Lap loss | 0.748 | 42.66 | 0.158 | 0.024 |
| w/ unsup. KP. | 0.746 | 44.86 | 0.138 | 0.023 |
| w/ concat | 0.737 | 46.38 | 0.144 | 0.025 |
| w/o inpaint | 0.744 | 43.95 | 0.140 | 0.022 |
| Full (VD) | **0.758** | **42.37** | **0.138** | **0.022** |

15 of Appendix. D.4, without the pre-training process, the identity similarity, image quality, and expression accuracy drop significantly. Besides, as shown in line 2, fine-tuning is also necessary to achieve a low AED. We suspect it is because the pre-trained I2P only learns to reconstruct the 3D avatar with the source expression, and it needs further updates to support face animation given the target expression. Besides, there is a domain gap between the image dataset and the video dataset, hence fine-tuning is necessary to achieve better visual quality. As for the I2P model scale, as shown in line 3 and 4, we found that 87M achieves significantly better image quality than 50M, while the performance difference between default settings to 200M is not obvious. In line 5, we found laplacian loss is necessary to improve motion-conditioned head animation.

**HTB-SR**   We test three settings on the HTB-SR model: (1): *w/ unsup. KP.*, which is similar to Face-vid2vid that jointly learns a predictor to extract unsupervised driving key points from the predicted head image; (2) *w/ concat*, which replaces the proposed alpha-blending-style fusion module in Eq.5 with a naive channel-wise concatenation of the head/torso/background feature maps; (3) *w/o inpaint*, which removes the KNN-based inpainting of the background image. The results are shown in Table 4. In line 1, we can observe that unsupervised key points lead to worse visual quality due to the instability of the extra predictor network. In line 2, we find that alpha-blending-style fusion is necessary to obtain a good identity preserving and image quality and eliminate the artifacts shown in Fig. 16 of Appendix. D.4. In line 3, we find removing the background inpainting process results in worse image quality.

## 5 CONCLUSION

In this paper, we propose a framework for one-shot and realistic 3D talking portrait synthesis, namely Real3D-Portrait. Our method simultaneously achieves accurate 3D avatar reconstruction and animation by designing a pre-trained large Image-to-plane model and a PNCC-conditioned motion adapter. Thanks to the proposed HTB-SR model, our method is also the first one-shot 3D method that could generate realistic video with natural torso movement and switchable background. Besides, with the introduction of a generic audio-to-motion model, our method is the first work that supports video/audio-driven applications. Extensive experiments demonstrate that our method surpasses state-of-the-art baselines from the perspective of identity preserving, visual quality, and audio-lip synchronization. Due to space limitations, we discuss limitations and future works in Appendix E.

## 6 ACKNOWLEDGMENTS

This work was supported in part by the National Natural Science Foundation of China under Grant No. 62222211 and National Key R&D Program of China under Grant No.2022ZD0162000.

## 7 ETHICS IMPACTS

Real3D-Portrait facilitates one-shot and realistic 3D talking portrait synthesis. With the development of talking face generation techniques, it is much easier to synthesize talking human portrait videos. Under appropriate usage, this technique could facilitate real-world applications like virtual idols and customer service, improving the user experience and making human life more convenient. However, the talking face generation method can be misused in deepfake-related usages, raising ethical concerns. We are highly motivated to handle these misusage problems. To this end, we plan to include several restrictions in the license of Real3D-Portrait. Specifically, (1) we will add visible watermarks to the video synthesized by Real3D-Portrait so that the public can easily tell the fakeness of the synthesized video. (2) The synthesized videos should only be used in educational or other legal usages (like online courses), and any abuse will take responsibility by tracking the method we come up with in the next point. (3) We will also inject an invisible watermark into the synthesized video to store the information of the video maker so that the video maker has to account for the potential risk raised by the synthesized video.

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

Table 5: The illustration of properties of different talking face generation methods.

| Method | One-shot? | 3D-Aware? | Natural Torso? | Switchable BG? | Driving Resource |
|---|---|---|---|---|---|
| Face-vid2vid (Wang et al., 2021) | ✓ | ✗ | ✗ | ✗ | video |
| HiDe-NeRF (Li et al., 2023a) | ✓ | ✓ | ✗ | N/A | video |
| OTAvatar (Ma et al., 2023) | ✓ | ✓ | ✗ | N/A | video |
| MakeItTalk (Zhou et al., 2020) | ✓ | ✗ | ✓ | ✗ | audio |
| PC-AVS (Zhou et al., 2021) | ✓ | ✗ | ✓ | ✗ | audio |
| RAD-NeRF (Tang et al., 2022) | ✗ | ✓ | ✓ | ✓ | audio |
| **Real3D-Portrait** (ours) | ✓ | ✓ | ✓ | ✓ | audio/video |

## A  COMPARSION BETWEEN DIFFERENT METHODS

Our method also holds the idea of *3D reconstruction and 3D animation* as (Li et al., 2023a) and (Li et al., 2023b) do, but further improves the 3D reconstruction power by proposing an image-to-plane (I2P) pretraining stage and enhances the face animation quality with motion adapter (MA), a PNCC-conditioned diff-plane predictor, hence achieves the goal of accurate 3D reconstruction and good animation quality.

The difference between our method and previous methods is obvious: Instead of previous end-to-end training, we propose a *pretrain-and-finetune* framework that simultaneously achieves the goals of accurate 3D reconstruction and stable face animation. To be specific, our Real3D-Portrait first distills 3D prior knowledge from a 3D GAN to pre-train an image-to-plane (I2P) model, then fine-tune the I2P model alongside a motion adapter (MA) on a video dataset to learn a dynamic motion-conditioned 3D talking face renderer; Besides, we are the first to consider natural torso movement and switchable background; Finally, we are the first work that achieves both of audio and video-driven applications. For better comparison, we list the property of our method and several state-of-the-art baselines in Table 5.

**Our difference from HiDe-NeRF and GOS-Avatar**   As HiDe-NeRF (Li et al., 2023a) and GOS-Avatar (Li et al., 2023b) are the most relevant baselines of Real3D-Portrait that belong to the predictor-based one-shot 3D talking face generation paradigm, we discuss our difference from them as follows. (1) We propose to pre-train the I2P model on an image multi-view dataset while the baselines don't; (2) Most importantly, we propose a motion adapter that predicts a motion diff-plane to directly morph the canonical tri-plane from the source expression into the target expression. By contrast, HiDe-NeRF learns a deformation field, which results in bad geometry and bad visual quality (please refer to the video demo at `https://real3dportrait.github. io/static/videos/Comparison_with_deformation.mp4` for better demonstration); and GOS-Avatar depends on an extra expression neutralization to the source image, which requires additional supervision signals and may cause information loss in the source image. (3) We propose a head-torso-background paradigm that individually models the head/torso/background segments, hence achieving realistic torso movement and overall good video naturalness. By contrast, the baselines don't consider the background and model the head-torso as a whole. (4) We propose a generic audio-to-motion model to support the audio-driven task, while the baselines only support the video-driven task, which limits their usage in real-world applications.

## B  ADDITIONAL NETWORK AND TRAINING DETAILS

In this appendix, we present the detailed network structure and training details of Real3D-Portrait.

### B.1  PRETRAINING IMAGE-TO-PLANE MODEL

We perform a *multi-view reconstruction* (Trevithick et al., 2023) task to effectively learn a feed-forward I2P model that maps the input image to a 3D tri-plane representation. Specifically, we adopt a pre-trained EG3D generator to yield tri-planes **P** of various synthesized persons from the latent space.

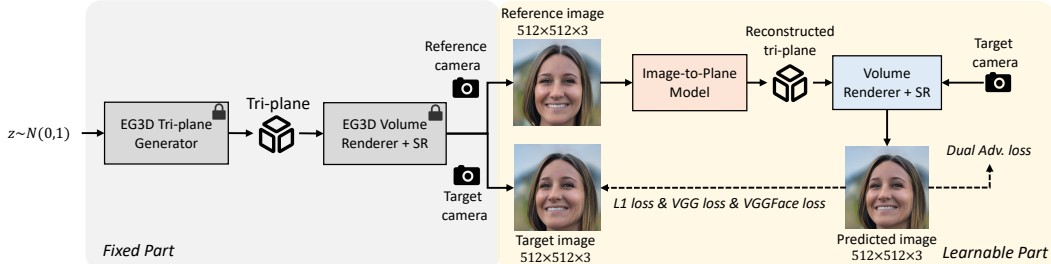

Figure 4: The pretraining process of I2P model in Sec. 3.1.

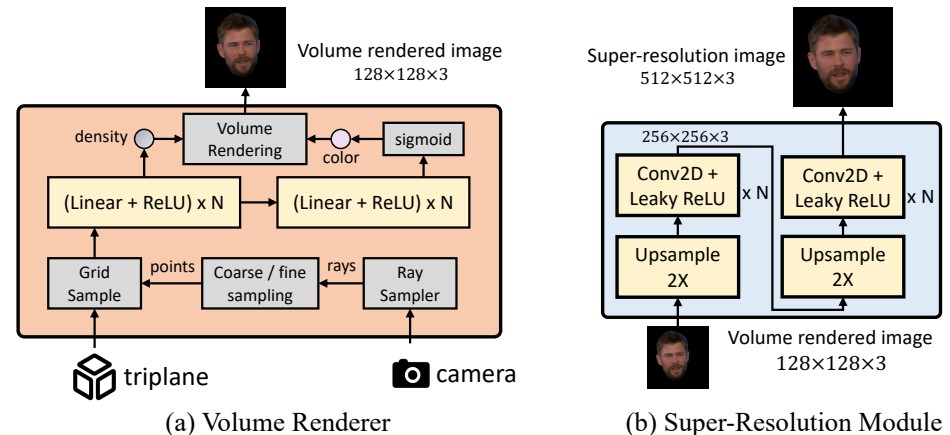

(a) Volume Renderer          (b) Super-Resolution Module

Figure 5: The network structure of volume renderer and naive super-resolution model used in the pretraining stage.

With the volume rendering technique, we could render images of the same person corresponding to the tri-plane $\mathbf{P}$ from an arbitrary viewpoint given the camera parameters $\mathbf{c}$. During training, to prepare the training data pair, we use an EG3D generator to generate tri-planes $\mathbf{P}$ of various identities, then use $\mathbf{P}$ to condition the volume renderer to synthesize two images ($\mathbf{I}_{\text{ref}}$ and $\mathbf{I}_{\text{mv}}$) of this identity from a reference camera $\mathbf{c}_{\text{ref}}$ and a multi-view camera $\mathbf{c}_{\text{mv}}$. Note that EG3D requires the camera to be sampled from a cycle with a fixed radius of 2.7, which is not desirable in talking face generation, where we want an appropriate portion between the head and torso. Hence, we relax the fixed-radius constraint of the camera so the camera distance can range from 2.4 to 5.0. As shown in Fig 4, our image-to-plane model takes the reference image $\mathbf{I}_{\text{ref}}$ as input to reconstruct a canonical tri-plane $\overline{\mathbf{P}}$ of the target person, then we volume renders the reconstructed tri-plane from the viewpoint of $\mathbf{c}_{\text{mv}}$ to render a multi-view image $\overline{\mathbf{I}}_{\text{mv}}$. Intuitively, we use the error between $\overline{\mathbf{I}}_{\text{mv}}$ and $\mathbf{I}_{\text{mv}}$ to provide a supervision signal to the image-to-plane model. Specifically, the training objective is as follows:

$$\mathcal{L} = \text{MSE}(\mathbf{I}_{\text{mv}}, \overline{\mathbf{I}}_{\text{mv}}) + \text{VGG}_{19}(\mathbf{I}_{\text{mv}}, \overline{\mathbf{I}}_{\text{mv}}) + \text{VGG}_{\text{Face}}(\mathbf{I}_{\text{mv}}, \overline{\mathbf{I}}_{\text{mv}}) + \text{DualAdv}(\overline{\mathbf{I}}_{\text{mv\_raw}}, \overline{\mathbf{I}}_{\text{mv}}) \quad (7)$$

where MSE is the mean-squared-error between $\overline{\mathbf{I}}_{\text{mv}}$ and $\mathbf{I}_{\text{mv}}$; $\text{VGG}_{19}$ and $\text{VGG}_{\text{Face}}$ is the perceptual loss empowered by a pre-trained VGG19 (Simonyan & Zisserman, 2014) and VGGFace (Parkhi et al., 2015) network; DualAdv denotes the dual discrimination proposed by EG3D (Chan et al., 2022), which could improve image fidelity and encourage keeping view consistency between the volume-rendered low-resolution image $\overline{\mathbf{I}}_{\text{mv\_raw}}$ and the super-solution image $\overline{\mathbf{I}}_{\text{raw}}$. We used the pre-trained Dual Discriminator provided by Chan et al. (2022) and finetuned it during the training process.

## B.2 OBTAINING PNCC FROM 3DMM COEFFICIENTS

We use the projected normalized coordinate code (PNCC) (Zhu et al., 2016; Li et al., 2023a) as the motion representation, which is an appearance-agnostic feature image that only related to the facial geometry, as the input condition to morph the 3D face. To be specific, PNCC can be formulated as:

$$\text{PNCC} = \text{Z-Buffer}(\text{Vertex}_{3D}(\mathbf{i}, \mathbf{e}), \text{NCC}), s.t. \text{Vertex}_{3D}(\mathbf{i}, \mathbf{e}) = \overline{\text{Vertex}_{3D}} + B_{\text{id}}\mathbf{i} + B_{\text{exp}}\mathbf{e} \quad (8)$$

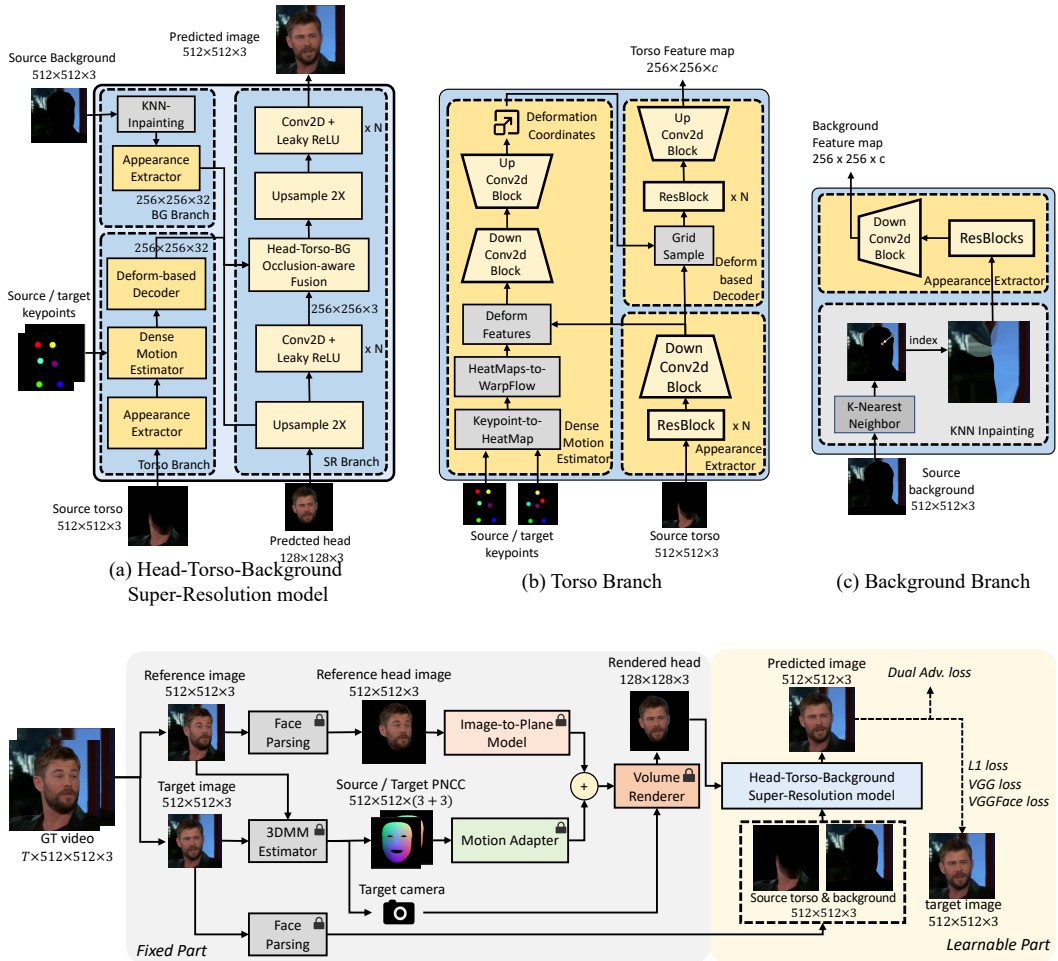

Figure 7: The training process of HTB-SR model in Sec. 3.3.

where NCC is the normalized coordinate code provided by Zhu et al. (2016) and acts as the colormap in the Z-Buffer (Phong, 1998) rendering process; $\text{Vertex}_{3D}$ is the vertex of a reconstructed 3DMM face that in the canonical space, which is determined by an 80-dimension identity code $\mathbf{i}$ and a 64-dimension expression code $\mathbf{e}$; $\overline{\text{Vertex}}_{3D}$, $B_{id}$, and $B_{exp}$ are the template shape, identity basis, and expression basis of a 3DMM model (Blanz & Vetter, 1999). This way, we could obtain PNCC from 3DMM identity/expression coefficients. Note that we could extract identity and expression coefficients from an image via 3DMM fitting. In the video-driven applications, we use the identity code from the source image and the expression code sequence from the driving video to construct the driving PNCC; as for the audio-driven applications, the required expression code sequence is predicted by the generic audio-to-motion model given the input audio.

## B.3 WARPING-BASED TORSO BRANCH

In our observation, it is uncommon for the torso part to rotate, and its dynamic can be regarded as nearly a joint translation together with the head part. This observation motivates us to model the torso part with a 2D warping-based renderer, which is computationally efficient and proven robust in complicated scenes (Siarohin et al., 2019). As shown in Fig. 6(b), our torso branch can be viewed as a warping-based Face-vid2vid (Wang et al., 2021) model that only renders the torso segment and is driven by predefined key points (instead of unsupervised keypoints (Wang et al., 2021) predicted from the image). To be specific, firstly, we obtain the predefined keypoint with:

$$\text{KP} = \text{IdxKP}(\mathbf{R} \cdot \text{Vertex}_{3D} + \mathbf{t}) \tag{9}$$

where Vertex$_{3D}$ is the 3DMM vertex reconstructed from identity and expression code defined in Eq. 1, $\mathbf{R}$ and $\mathbf{t}$ is the rotation matrix and translation of the extracted camera, IdxKP denotes select 68 facial keypoints (Bulat & Tzimiropoulos, 2017) from the 3DMM vertex. The key points are fed into the deformation motion estimator (DME) shown in Fig. 6(b) to predict the deformation pixel coordinates of the torso segment. Then, we grid sample from the source torso appearance feature map with the predicted deformation field to obtain the warped torso feature map. The overall warping-based torso rendering can be expressed as:

$$\mathbf{F}_{torso} = \text{DBD}(\text{TAE}(\mathbf{I}_{torso}), \text{DME}(\text{KP}_{src}, \text{KP}_{tgt})) \tag{10}$$

where TAE, DME, and DBD are the torso appearance encoder, dense motion estimator, and deformation-based decoder in Fig. 6(a). One could refer to (Wang et al., 2021) for more details about these modules.

## B.4 Background Branch with KNN-based Inpainting

The biggest challenge to achieving a realistic background is generating the pixels occupied by the foreground (i.e., the person) in the source image. To this end, we first adopt a K-nearest-neighbor-based inpainting method to preprocess the background segment of the source image. Specifically, for each foreground pixel, we find its nearest neighbor that belongs to the background segment, then fill the foreground pixels with the color of their nearest background pixels. Once we obtained an inpainted background image, as shown in Fig. 6(c), we fed it into a VGG-style appearance extractor to extract the background feature map. Note that since we individually model the background, we support switching backgrounds during inference. With the previous volume rendering, we obtain the low-resolution head image; with the torso branch and background branch, we obtain the torso and background feature map. Then, we use a super-resolution branch to integrate these three segments and generate a $512{\times}512$ composite image, which is shown in Fig. 6(a).

## B.5 Obtaining the Head and Torso Occlusion Mask

In Eq. 5, we propose to obtain the final talking portrait image with an alpha-blending-style fusion of head/torso/background feature maps. In this section, we introduce how to obtain the occlusion mask $\mathbf{M}_{head}$ and $\mathbf{M}_{torso}$ required in Eq. 5 by utilizing the nature of NeRF-based head and warping-baed torso rendering module.

As for the NeRF-based head segment, following AD-NeRF (Guo et al., 2021), we could obtain a head mask $\mathbf{M}_{head}$ by volume rendering:

$$\mathbf{M}_{head} = \int_{t_n}^{t_f} \sigma(\mathbf{r}(t)) \cdot \exp\left(-\int_{t_n}^{t} \sigma(\mathbf{r}(s))ds\right) dt, \tag{11}$$

where $\sigma$ is the density predicted by the NeRF, $\mathbf{r}$ and $t$ are the ray and ray marching depth of the volume rendering technique. $t_n$ and $t_f$ denote the nearest and farthest point of the ray.

As for the warping-based torso segment, following Face-vid2vid (Wang et al., 2021), when designing the dense motion estimator shown in Figure 6(a), apart from predicting the dense motion flow, the network also predicts a 1-dimension occlusion mask for the torso $\mathbf{M}_{torso}$.

## B.6 Detailed Structure of Audio-to-Motion Model

We illustrate the detailed network structure of the A2M model in Fig.8. The overall model consists of two generative models: a VAE as the main structure and a Flow-based model as the enhanced prior of VAE. We use WavNet as the backbone of the encoder, decoder, and flow-based prior. 3DMM expression code, a 64-dimension vector, is chosen as the motion representation to be predicted, so the in-out dimension of the A2M model is $T \times 64$, where $T$ is the time dimension. During inference, it is convenient to obtain PNCC via the Z-Buffer algorithm given the predicted 3DMM expression code, as illustrated in Appendix B.2.

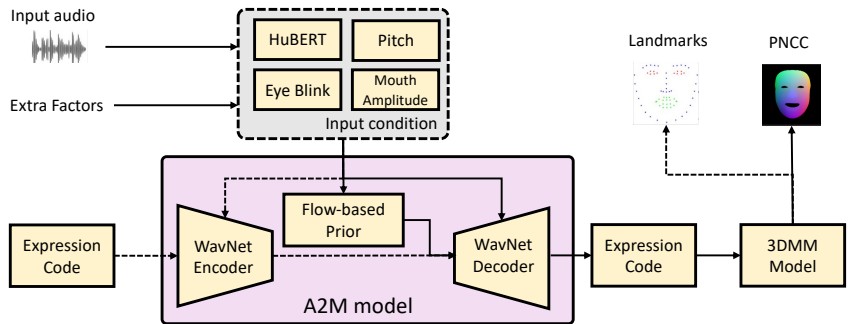

Figure 8: The network structure of A2M model. Dotted lines denote the processes that are only executed during the training phase in Sec. 3.4.

## C DETAILED MODEL CONFIGURATION

### C.1 MODEL CONFIGURATION

We provide detailed hyper-parameter settings about the model configuration in Table 6.

Table 6: Model Configuration

| | Hyper-parameter | Value |
|---|---|---|
| I2P Model (87M) | ViT Branch - Patch Size | 3 |
| | ViT Branch - Patch Emb Channel | 1024 |
| | ViT Branch - Attention Blocks | 6 |
| | ViT Branch - MLP Layers per Block | 2 |
| | ViT Branch - Attention Block Channel | 1024 |
| | VGG Branch - Conv2D Layers | 10 |
| | VGG Branch - Conv2D Channels | 256 |
| | Final Conv2D Layers | 4 |
| | Output Feature Map Size | $256 \times 256 \times 32 \times 3$ |
| Motion Adapter (5.7M) | Patch Size | 4 |
| | Patch Emb Channels | [32, 64, 160, 256] |
| | Norm Type | LayerNorm |
| | Self-Attention Block | 4 |
| | MLP Layers per Block | 2 |
| | Attention Heads | [1, 2, 5, 8] |
| | Drop Path Rate | 0.1 |
| | Discrimnator Dropout Rate | 0.25 |
| Volume Renderer (0.004M) | MLP Layers | 2 |
| | MLP Channels | 64 |
| HTB-SR Model (50M) | Torso Branch - TAE - Conv2D/3D Layers | 3 + 6 |
| | Torso Branch - MFE - Conv3D Layers | 13 |
| | Torso Branch - DBD - Conv2D Layers | 9 |
| | BG Branch - KNN number of neightbors | 1 |
| | BG Branch - Appearance Encoder Conv2D layers | 3 |
| | Head-Torso-BG Fusing Conv2D Layers | 6 |
| | Conv2D/3D Kernel | 3 |
| A2M Model (10M) | Encoder WavNet Layers | 8 |
| | Decoder WavNet Layers | 4 |
| | Encoder/Decoder Conv1D Kernel | 5 |
| | Encoder/Decoder Conv1D Channel Size | 192 |
| | Latent Size | 16 |
| | Prior Flow Layers | 4 |
| | Prior Flow Conv1D Kernel | 3 |
| | Prior Flow Conv1D Channel Size | 64 |

## C.2 TRAINING DETAILS.

All training processes of Real3D-Portrait are performed on 8 NVIDIA A100 GPUs. As for the renderer, we first pre-train the I2P model for 250,000 steps, which takes about 72 hours; then we train the motion adapter for 200,000 steps, which takes about 60 hours; finally, we train the HTB-SR model for 200,000 steps, which takes about 30 hours. As for the A2M model, we train it for 100,000 steps, which takes about 16 hours.

## D ADDITIONAL EXPERIMENTS

### D.1 EVALUATION DETAILS

In this section, we illustrate details for collecting the data for evaluation.

As for collecting the source image and driving audio/video, there are three groups of evaluated data: (1) for the Same-Identity Reenactment, we randomly chose 100 videos in our preserved validation split of CelebV-HQ. (2) for the Cross-Identity Reenactment, we use the first frame of the previously selected videos to obtain 100 identities, then use the expression-pose sequence from a randomly selected video to construct the cross-identity reenactment data pair. (3) for the audio-driven scenario, we use 10 out-of-domain images downloaded from the internet (which are exactly the 10 identities shown in the demo video) and choose 10 audios from different languages to form the data pair (so each method has 100 videos as the test samples).

As for choosing the camera pose, since the prediction of the head pose is not our main interest, we devise a naive strategy to obtain the head pose sequence from GT videos of CelebV-HQ. To be specific, (1) in the same-identity setting, the driving head pose is exactly the same as the GT video of the source image (source image is the first frame of the test video); (2) in the cross-identity setting, the driving head pose is extracted from the driving video; (3) then, in the audio-driven setting, most importantly, we first estimate the head pose in the source image, then we query 10 videos in the CelebV-HQ that has nearest distance between its first frame's head pose and the source image. Once the candidate videos that provide head pose are sampled, we randomly choose one of them to drive the source image.

### D.2 USER STUDY SETTING

We selected ten audio/video clips and ten identities to construct 100 talking portrait video samples for each audio/video-driven method. We involved 20 participants in rating each video. We perform the MOS evaluations from the aspect of identity preservation, visual quality, and audio-lip synchronization. Each tester is asked to evaluate the subjective score of a video on a 1-5 Likert scale. For identity preservation, we tell the participants to *"only focus on the similarity between the identity in the source image and the video"*; for visual quality, we tell the participants to *"focus on the overall visual quality, including the image fidelity and smooth transition between adjacent frames"*; as for audio-lip synchronization, we tell the participants to *"only focus on the semantic-level audio-lip synchronization, and ignores the visual quality"*.

### D.3 ADDITIONAL QUALITATIVE RESULTS

**Overall Comparison.** We provide a qualitative comparison with all video/audio-driven baselines in Fig. 9 and Fig.10.

**PNCC-conditioned Face Animation.** We illustrate how PNCC animates the 3D avatar in Fig. 11. The first column is the input head image of the image-to-plane model, and the second column is the input driving PNCC of the motion adapter model. The third column is the low-resolution rendered image ($128 \times 128$) produced via volume rendering, and the fourth column is the corresponding depth image, which helps visualize the 3D geometry of the modeled 3D avatar. The fifth column shows the high-resolution rendered image ($512 \times 512$) processed by the naive SR module used when training the motion adapter.

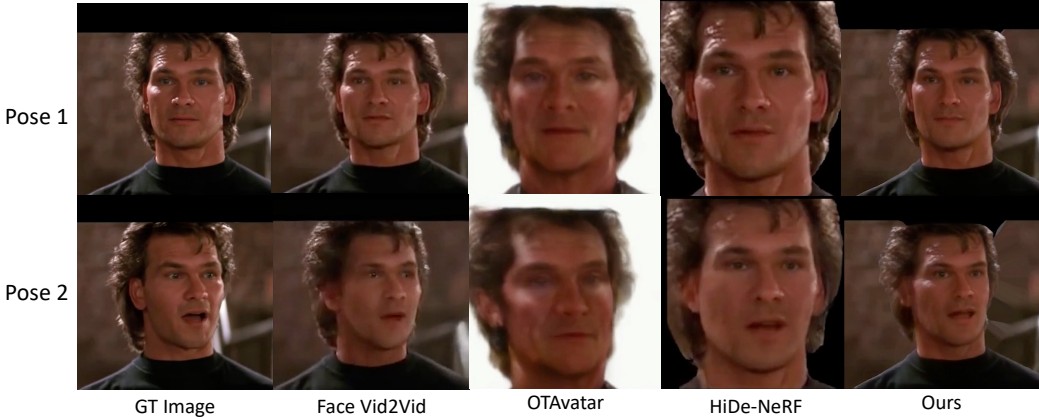

Pose 1

Pose 2

GT Image    Face Vid2Vid    OTAvatar    HiDe-NeRF    Ours

Figure 9: Qualitative Comparison with video-driven baselines. We recommend the reader refer to the demo video at `https://real3dportrait.github.io/static/videos/Comparison_with_VD_baselines.mp4` for clear comparison. In this figure, we can see that (1) Face-vid2vid degrades at a large head pose; (2) OTAvatar cannot produce an identity-preserving result; (3) HiDe-NeRF produces texture jittering artifacts given different poses; and (4) our method could produce identity-preserving and realistic results.

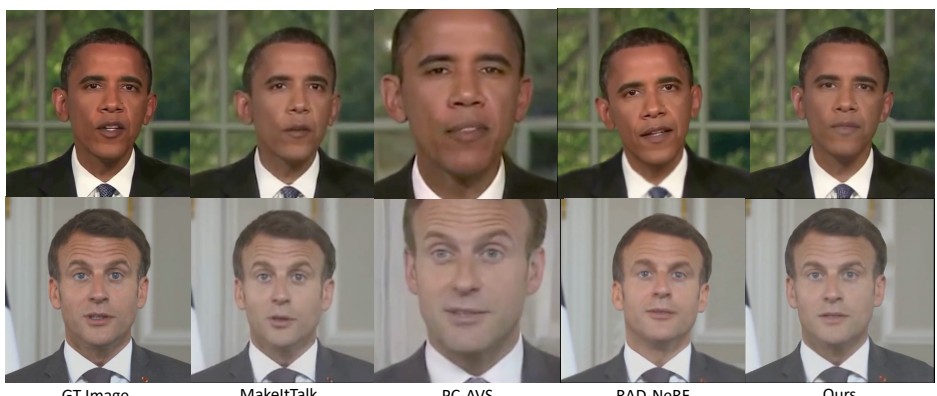

GT Image    MakeItTalk    PC-AVS    RAD-NeRF    Ours

Figure 10: Qualitative Comparison with audio-driven baselines. We recommend the reader refer to the demo video at `https://real3dportrait.github.io/static/videos/Comparison_with_AD_baselines.mp4` for clear comparison. In this figure, we can see that (1) MakeItTalk tends to produce cartoon-style results; (2) PC-AVS only renders the cropped face part and fails to achieve a good identity similarity; (3) our method and RAD-NeRF could generate identity-preserving and realistic talking portraits.

**Realistic Torso Movement.** As shown in Fig. 12, with the warping-based torso branch in the HTB-SR model, our method could generate realistic torso segments given large and critical head poses.

**Switchable Background.** As shown in Fig. 13, with the background branch in the HTB-SR model, our method supports switching background during inference.

**Audio-Lip Synchronization.** As shown in Fig. 14, with the generic audio-to-motion model, our method achieves audio-lip synchronization in the audio-driven scenario.

D.4    ADDITIONAL ABLATION STUDIES

**Pretraining I2P Model on a Multi-view Image Dataset** As shown in Fig. 15, without the pretraining stage and learning the I2P model from scratch on the video dataset leads to degraded

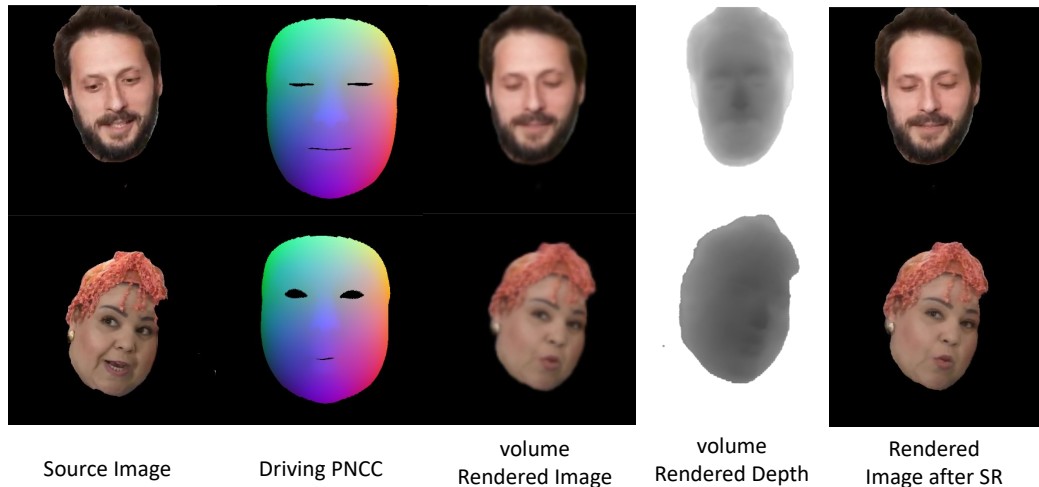

| Source Image | Driving PNCC | volume Rendered Image | volume Rendered Depth | Rendered Image after SR |

Figure 11: Illustration of how PNCC animates the 3D head.

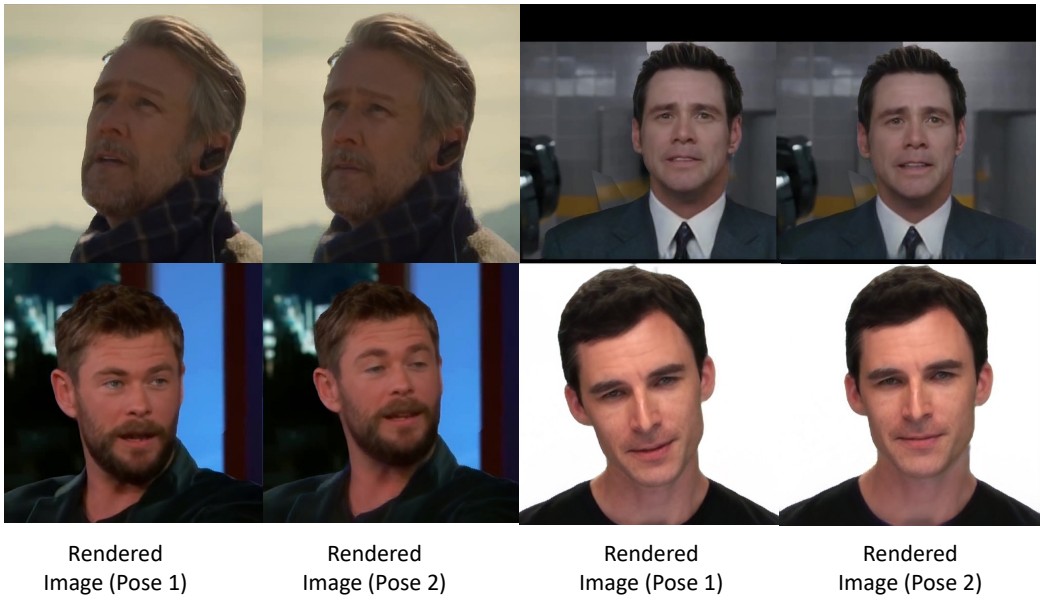

| Rendered Image (Pose 1) | Rendered Image (Pose 2) | Rendered Image (Pose 1) | Rendered Image (Pose 2) |

Figure 12: Demonstration that our method could generate realistic torso segments given different head poses.

image quality and inaccurate facial animation. By contrast, with the pre-trained I2P model, we achieve high image fidelity, good identity-preserving ability, and accurate face motion control.

**Alpha-Blending-Style Fusion in HTB-SR Model** We compare results with/without the proposed head-torso-background alpha-blending fusion in Fig. 16. (1) As can be seen in the first image, without alpha-blending, i.e., directly concatenating the feature maps of head/torso/background results in a hollow artifact in the hair region. We suspect that it is caused by the unrestricted spatial information changing between these three semantic feature maps. For instance, in this case, the features from the background image override the information from the head image. By contrast, by using the face mask, we could eliminate the background information within the head region, which addresses the hollow artifact (as shown in the second image in Fig. 16). (2) Besides, as seen in the third image, without a mask to restrain the spatial feature fusion, the head-torso boundary region (marked by a

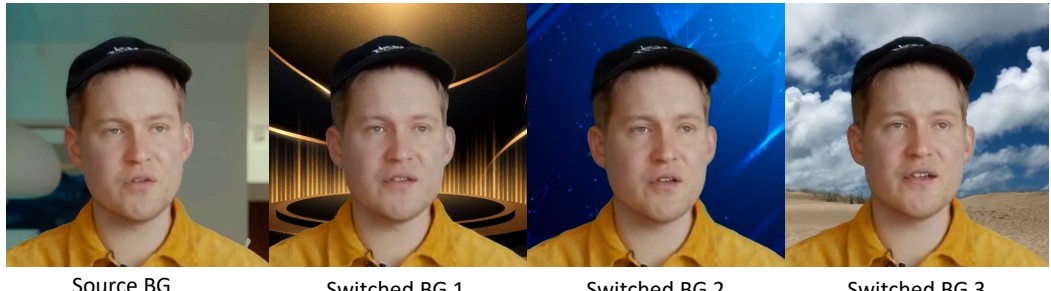

Source BG     Switched BG 1     Switched BG 2     Switched BG 3

Figure 13: Demonstration that our method supports switchable background.

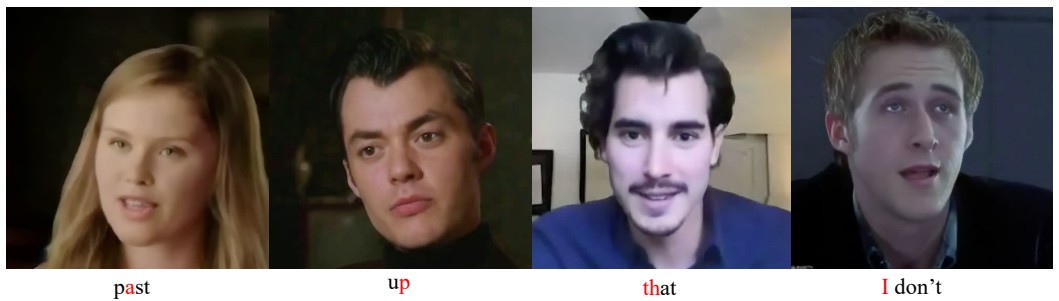

past     up     that     I don't

Figure 14: Demonstration that our generic audio-to-motion model could generate accurate lip motion.

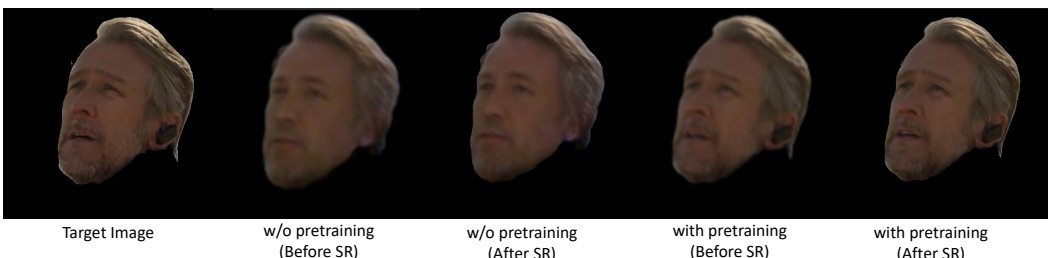

Target Image    w/o pretraining (Before SR)    w/o pretraining (After SR)    with pretraining (Before SR)    with pretraining (After SR)

Figure 15: Comparison between with or without the pretraining process of I2P model as introduced in Sec. 3.1.

red rectangle) seems unrealistic and blurry. By contrast, with the proposed alpha-blending fusion technique, we could generate realistic and sharp results in the boundary region.

### D.5 ADDITIONAL QUANTITATIVE COMPARISON WITH RECENT 2D BASELINES

We additionally compare with several remarkable 2D baselines, such as DaGAN (Hong et al., 2022a), TPS (Zhao & Zhang, 2022), DPE (Pang et al., 2023), LIA (Wang et al., 2022), and MCNet (Hong & Xu, 2023), and provide a demo video for qualitative comparison in `https://real3dportrait.github.io/static/videos/Comparison_with_5_additional_VD_baselines.mp4`. We also provide quantitative comparison in Table 7. We can see that our method achieves the best performance in terms of CSIM, FID, AED and APD.

## E LIMITATIONS AND FUTURE WORK

Firstly, due to the absence of large-posed images in the training data, our method fails to generate images under large head poses like side views. We plan to address this problem by introducing

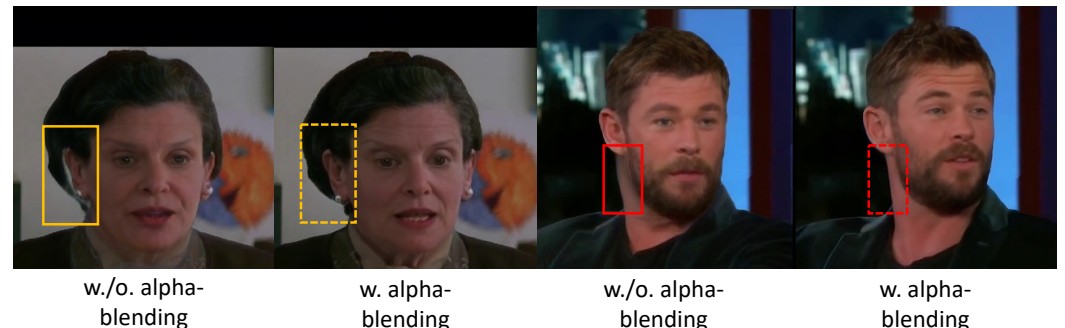

| w./o. alpha-blending | w. alpha-blending | w./o. alpha-blending | w. alpha-blending |

Figure 16: Comparison between with or without alpha-blending-style head-torso-background fusion as introduced in Eq. 5. We use the rectangle to point out the artifacts caused by not using the alpha-blending fusion and use the rectangle with dotted to show that using alpha-blending fusion could address these problems.

Table 7: Additional comparison with 2D VD baselines.

| Methods | CSIM↑ | FID↓ | AED↓ | APD↓ |
|---|---|---|---|---|
| TPS (Zhao & Zhang, 2022) | 0.745 | 43.54 | 0.137 | 0.028 |
| DPE (Pang et al., 2023) | 0.731 | 44.07 | 0.135 | 0.021 |
| MCNet (Hong & Xu, 2023) | 0.758 | 42.49 | 0.136 | 0.030 |
| Real3D-Portrait | **0.764** | **41.58** | **0.129** | **0.017** |

more large-posed data and improving the tri-plane 3D representation. Secondly, the image quality can be improved by introducing more high-fidelity training data and more delicately designed networks. Thirdly, a few-shot in-context-learning 3D talking face method is desirable for better identity preservation and visual quality. Finally, though we have achieved generally high-quality realistic talking portrait results, one of the limitations is that the inpainted background image could leak unnaturalness when the talking person is doing a large pose motion. We believe the introduced naive KNN-based background inpainting method is to be blamed. Since generating a realistic background is of high importance to ensure the realism of the final video, we plan to upgrade the background inpainting method with a more advanced neural network-based system, such as LAMA (Suvorov et al., 2021). An alternative might be learning the background inpainting module within the HTB-SR model in an end-to-end manner. We leave this for future work.

