# OpenReview forum: "Real3D-Portrait: One-shot Realistic 3D Talking Portrait Synthesis"
_ICLR.cc/2024/Conference — ICLR 2024 spotlight_

### Official Review · Reviewer_Kntd · 2023-10-31

**Soundness:** 3 good
**Presentation:** 3 good
**Contribution:** 3 good
**Rating:** 8
**Confidence:** 4

**Summary:**

The paper proposes a method for realistic one-shot 3D talking face generation driven by videos and audio. Given an input source image, the method first reconstructs a canonical 3D face in the tri-plane representation with an Image-to-Plane (I2P) model. The I2P comprises 1. a ViT branch with a stack of SegFormer blocks to handle the canonicalization and a VGG branch to capture the high-frequency appearance features. The driving motion is represented by the projected normalized coordinate code (PNCC) [Zhu et al. 2016]. Instead of using the deformation field to animate as in previous papers, the method proposes to have a motion adapter composed of a shallow SegFormer to output the residual motion diff-plane given the PNCC. The sum of the canonical tri-plane and the motion diff-plane are volume-rendered and upsampled by a super-resolution module to end up with the source image driven by the given motion. Lastly, the background and the torso are composited with the Head-Torso-Background supersampling model. The driving PNCC can come from audio using an audio-to-motion model using a flow-enhanced VAE or video by fitting 3DMM to the reference video. The method is compared against Face-vid2vid [Wang et al. 2021], OT-Avatar [Ma et al. 2023], HiDe-NeRF [Li et al. 2023a] for the video-driven animation, and MakeItTalk [Zhou et al. 2020], PC-AVS [Zhou et al. 2021], and RAD-NeRF [Tang et al. 2022] for the audio-driven animation.

**Strengths:**

Out of the contributions that the authors claim, the motion adapter generating the residual motion diff-plane to animate the canonical tri-plane given the target PNCC seems to be the most interesting part of this work.

**Weaknesses:**

Some design decisions may not be well-supported. See my questions.

**Questions:**

Can authors clarify which 3DMM they use? Is it BFM2009 as referred from section 3.4, or is it [Blanz and Vetter 1999] as referred from B.2? Note that while they both use PCA, their bases are not compatible because they are computed from different data.

(Related to my question above)
In 3.4, authors say:
> 3DMM basis is a nonhomogeneous linear equation and has multiple solutions (e.g., BFM2009)...

At least for PCA-based 3DMMs like BFM2009 and [Blanz and Vetter 1999], this is incorrect. Because PCA bases are orthogonal, a unique least squares solution can be found by simply multiplying the principal components and the data (target 3D mesh vertex positions in this case). Because of this misunderstanding, the paper may have an over-complex setup in its audio-to-motion model.

How does the I2P model handle an input image with a non-neutral face? I.e., if the model is given an image with an extreme facial expression as the source image, what happens? Have authors thought about canonicalization in this respect? I believe a concurrent work, "Generalizable One-shot Neural Head Avatar," canonicalizes the input expression as well. Perhaps this should be clarified as a limitation.

Can authors actually show results if the deformation field is used instead of the residual motion diff-plane? The paper just says "bad quality," but this is never properly compared. The comparison video says the main issue with HiDe-NeRF (which uses the deformation field) is the temporal jittering. However, the temporal jittering is mitigated in this paper by the temporal Laplacian loss, not by the residual motion diff-plane.

Can authors provide results from multiple views to show how good the 3D reconstruction is?

Can authors consider comparisons to Next3D [Sun et al. 2023]? I understand that Next3D is an animatable 3D GAN model requiring a GAN inversion to reproduce an identity. A discussion on a concurrent work, "Generalizable One-shot Neural Head Avatar," is not required but much appreciated to understand the novelty of this work.

(minor typo)
3.1 Network Design
> comprises a stack of SegFormer bock

bock -> block

---

> ### Author Response · Authors · 2023-11-16
> **Author Response to Reviewer Kntd (Part 1/3)**
>
> Dear reviewer, we sincerely appreciate your careful and accurate understanding of our approach. We hope our response fully resolves your concerns.
>
> >  Q1: Can authors clarify which 3DMM they use? In 3.4, authors say: "3DMM basis is a nonhomogeneous linear equation and has multiple solutions". At least for PCA-based 3DMMs like BFM2009 and [Blanz and Vetter 1999], this is incorrect.  Because of this misunderstanding, the paper may have an over-complex setup in its audio-to-motion model.
>
> - A1: We use BFM2009 in all our experiments, though we have also cited [Blanz and Vetter 1999] in B.2. Following your comment, we checked the details in BFM2009 and found that our previous statement "3DMM basis is a nonhomogeneous linear equation and multiple solutions" is incorrect since the expression bases are orthogonal. Due to this problem, it is not necessary for the audio-to-motion model to avoid the MSE loss of expression (in the previous version to prevent over-smoothness, we designed an MSE loss of landmark). To handle this mistake, we have updated the discussion in the corresponding part of the revised manuscript. Specifically, **in the blue-colored part in Section 3.4 of the revised manuscript**, we replace the incorrect statement with:
>
>   - >  [Revised Part 1] We choose BFM 2009 \citep{bfm09} as the 3DMM model. Since all expression bases are orthogonal, given the same identity code, the reconstructed 3D face meshes in a video are uniquely determined by the expression code. Hence an L2 error on the expression code, L_ExpRecon, is feasible to be the reconstruction term in training the VAE. To encourage the model to better reconstruct the facial landmark (instead of only the 3DMM parameters), we additionally introduce the L2 reconstruction error of 468 key points of the reconstructed 3DMM vertex, L_LdmRecon, as an auxiliary supervision signal. The training loss of the generic audio-to-motion model is as follows:
>
> - Further, to fully investigate the influence of using MSE loss on expression and landmark, we compare three settings: (1) only use Expression MSE, (2) only use Landmark MSE, and (3) use both Expression / Landmark MSE. We compute the reconstructed landmark L1 error and SyncScore, and the results are shown as follows, we found that using Expression MSE loss alone achieves slightly better reconstruction performance than only using landmark MSE loss but leads to a slightly degraded sync score. we suspect that it is because for our model that predicts the expression code given audio, updating on an Expression MSE is straightforward and easier to optimize than on a Landmark MSE so the model would more focus on the reconstruction. Another finding is that combined usage of these two losses achieves the best performance in terms of reconstruction and sync score. Therefore, in the revised manuscript, we turn to using a combination of Expression MSE and Landmark MSE as the reconstruction loss of the VAE. We sincerely thank you for your careful reading and in-depth understanding of 3DMM which helps improve the correctness and soundness of this paper.
>    - | Loss Setting                                    | L1 error on pred. Ldms $\downarrow$ | Sync Score  $\uparrow$|
> | ----------------------------------------------- | --------------------------- | ---------- |
> | MSE_Landmark (in the original manuscript)       | 0.0113                      | 6.565      |
> | MSE_Expression                                  | 0.0104                      | 6.438      |
> | MSE_Expression + MSE_Landmark (revised version) | $\textbf{0.0102}$                     | $\textbf{6.617} $     |

---

> > ### Author Response · Authors · 2023-11-16
> > **Author Response to Reviewer Kntd (Part 2/3)**
> >
> > > Q2: (1) How does the I2P model handle an input image with a non-neutral face? (2)  Have authors thought about canonicalization in this respect? Discuss with a concurrent work, "Generalizable One-shot Neural Head Avatar," which canonicalizes the input expression.
> >
> > - A2: (1) Thanks for pointing out this problem! We acknowledge your concerns that if the motion adapter is only provided with the target PNCC, it cannot work well with the I2P model which could have arbitrary expression in the cano_planes. We apologize for not clarifying this issue in the previous manuscript. Actually, our key motivation in inventing the motion adapter is to directly perform the source-to-target transform in the cano_planes (skipping an intermediate expression canonicalization process that is required by GOS_Avatar's source-neutral-target pipeline). In practice, instead of only feeding the target PNCC into the motion adapter, the input is a channel-wise concated source-target PNCC image of shape $(C_{src}+C_{tgt})\times H\times W$. We sincerely thank for your insightful comments and we have updated the missed technical details of the motion adapter in the revised version (**Please refer to the blue-colored part in Section 3.2 of the revised manuscript**). In QA3 and QA4, we will provide two demos that are helpful for a better demonstration that our I2P+MA design could well handle source images of arbitrary expression. See [this link](https://real3dportrait.github.io/static/videos/Comparison_with_deformation.mp4) for the effectiveness of the motion adapter and [this link](https://real3dportrait.github.io/static/videos/I2P_multi_view_synthesis.mp4) for the performance of the I2P model.
> > - (2) As for "think about canonicalization in this process" and discussion on the difference with "Generalizable One-shot Neural Head Avatar" (GOS-Avatar).  To edit the expression in a non-neutral face image, it is a common practice to convert it into a neutral expression first, then perform a neutral-to-target expression (like os_avatar does). However, this two-stage pipeline (1) requires additional supervision signals (e.g., os_avatar adopts a 3DMM rendered neutral face as the intermediate target for the network to predict) and (2) may suffer from information loss due to the source-to-neutral-to-target process (e.g, for some OOD face, the expression neutralization cannot produce satisfactory neutral face and there may be texture detail information loss). Based on these insights, during the system designing process, we think a straightforward source-to-target method should perform more robustly and efficiently, which is exactly what our motion adapter does, and the demo in [this link](https://real3dportrait.github.io/static/videos/Comparison_with_deformation.mp4) will demonstrate our insight.  Thanks for your valuable comments that helped us better highlight the novelty, we have additionally discussed the difference between our method and GOS-Avatar in the revised version. (Please refer to the blue-colored part in Appendix A of the revised manuscript).
> >
> >
> > > Q3: Can authors actually show results if the deformation field is used instead of the residual motion diff-plane? The paper just says "bad quality," but this is never properly compared.
> > - A3: Thanks for pointing out this problem! Actually, "bad quality" denotes "bad quality of the reconstructed 3D geometry", which results in worse visual quality. Following your suggestion, we trained a model that only replace the real3d-portrait's Motion adapter with a deformation field in HiDe-NeRF to achieve face animation. To be more intuitive, we provide a [demo](https://real3dportrait.github.io/static/videos/Comparison_with_deformation.mp4) that visualizes the depth and color image of this deform-based model and our motion adapter-based model when driven by audio. Please kindly refer to [this link](https://real3dportrait.github.io/static/videos/Comparison_with_deformation.mp4) for viewing. As shown in the depth image, we can observe that instead of learning a 3D head mesh, the reconstructed 3D structure of the deformation field is more like a surface (i.e., only the most-frontend voxels contribute to final results), which is ill-posed and should account for the worse visual quality of this deformation-based system. By contrast, we can see that our I2P could normally produce a 3d head mesh, and then the MA only morphs the minimal parts of geometry corresponding to the input PNCC. As discussed above, we have shown the superiority of our MA over the deformation field for face animation. We also add this discussion in the blue-colored part in Appendix A of the revised manuscript. Thanks for your helpful suggestion that improves the soundness of this paper.

---

> > > ### Comment · Reviewer_Kntd · 2023-11-21
> > >
> > > I2P and expression canonicalization:
> > > The demo with non-neutral source images looks good. I think I now understand the setup can undo the expressions in the source because the PNCC is obtained from the source, and fed into the motion adapter along with the PNCC of the driving image.
> > > I see that the eq. 2 is also updated. Why are the authors not clarifying this? This is a critical technical update from the original. The figures also need updates with the correct inputs to the motion adapter.
> > >
> > > Deformation field vs. residual motion diff-plane:
> > > The demo looks good.

---

> ### Author Response · Authors · 2023-11-16
> **Author Response to Reviewer Kntd (Part 3/3)**
>
> > Q4: Can authors provide results from multiple views to show how good the 3D reconstruction is?
> - A4: Sure, please refer to [this link](https://real3dportrait.github.io/static/videos/I2P_multi_view_synthesis.mp4) for a demo of multi-view synthesis produced by our I2P model (Which is extracted from the Real3D-Portrait's final checkpoint). We can observe that the I2P could reconstruct accurate 3D shape and texture given the input image.
>
>
> > Q5: (1) Comparison to Next3D. (2) Also, add discussion on "Generalizable One-shot Neural Head Avatar" to help understand the novelty of this work.
>
> - A5: (1) As for the Next3D, we provide a comparison video in [this link](https://real3dportrait.github.io/static/videos/Comparison_with_Next3D.mp4). Note that since Next3D has not released their GAN inversion code, we reproduce a Next3D + PTI to achieve face reenactment by combining the official Next3D repo and an unofficial PTI repository. In the demo video, we can see that our method has the following advantages: rendering uncropped full frames, natural torso, better identity preservation, more accurate expression control, and more efficient (ours is one-shot while PTI requires 1000 iterations of tuning). As for qualitative results, we found Next3D+PTI is worse than ours in terms of CSIM, FID, AED, and APD.
>
>   -  | Method     | CSIM$\uparrow$   | FID$\downarrow$  | AED$\downarrow$  | APD$\downarrow$  |
>   | ---------- | ---------------- | ---------------- | ---------------- | ---------------- |
>   | Next3D+PTI | 0.645            | 58.66            | 0.183            | 0.039            |
>   | Ours       | $\mathbf{0.758}$ | $\mathbf{42.37}$ | $\mathbf{0.138}$ | $\mathbf{0.022}$ |
>
> - (2) As for the discussion on GOS-Avatar, as we discussed in $\mathbf{A2}$, our most related improvement is to design a motion adapter that efficiently performs the "source-to-target" transform, instead of GOS-Avatar's "source-to-neutral-to-target" transform that requires additional supervision signals and may bring information loss to the model (We suspect this might be reason that GOS-Avatar has to train an additional appearance branch to help the model to "remind" these texture details). Besides, we adopt a pretraining process to the I2P model, which is not performed by GOS-Avatar. Further, we are the first to consider the head-torso-background individual modeling, which is necessary to achieve realistic video results. Finally, we are also the first work that supports the end-to-end audio-driven task. We have added a discussion on the difference between our method and GOS-Avatar in the blue-colored part in Appendix A of the revised manuscript.
>
>
>
> > Q6: (minor typo) "3.1 Network Design comprises a stack of SegFormer bock."  bock -> block
>
> - A6: Thanks for your careful reading of this manuscript! We have fixed this typo in the revised manuscript accordingly.s
>
>
> ## Summary
> Following your comments, we have modified the manuscript and we found the correctness and soundness of the paper have been improved. Again, we thank the reviewer for the insightful review and positive recommendation for our paper.

---

> > ### Comment · Reviewer_Kntd · 2023-11-21
> >
> > Thanks for providing the evidence for the 3D reconstruction quality, comparison to Next3D, and a discussion on a concurrent work GOS-Avatar. They all look good.

---

> ### Comment · Reviewer_Kntd · 2023-11-21
>
> The revision on the 3DMM and the A2M model looks good to me.
>
> I honestly think the expression loss should be sufficient because the landmark positions can be linearly computed from the expression coefficients. But as long as the authors do not claim a huge technical contribution here, this is minor.

---

> ### Author Response · Authors · 2023-11-21
> **Thanks for your acknowledgment on our  rebuttal**
>
> Dear reviewer Kntd,
>
> Thanks for your acknowledgment on our  rebuttal. To answer the remaining problems:
>
> (1) About the clarifying the updated Eq.2. The Eq.2 is updated according to our clarification on the input feature of Motion adapter (as discussed in [Revised Part 2] in the revised PDF): we exactly feed the concatenation of source PNCC and driving (target) PNCC into motion adapter, so the input condition of motion adapter is (PNCC_drv, PNCC_src) instead of only (PNCC_drv). By perceiving the information of source expression and target expression, the MA could learn the source-to-target expression transformation. Thanks for pointing out this update, and we have also blue-colored the eq.2 to highlight the difference.
>
> (2) As for update the figure. We have updated the figure 1,2,3,4 in the manuscript with the correct inputs to the motion adapter.
>
> Again, we'd like to thank you for your precious time and valuable comments, which have improved the manuscript by a large margin.

---

> > ### Comment · Reviewer_Kntd · 2023-11-21
> >
> > Please clarify all updates to the revisions. I see figures 1, 2, and 4, not just figure 3,  are updated to denote the PNCC from the source going into the motion adapter.
> >
> > The authors have addressed all of my concerns. I am updating my score based on the revision.

---

> > > ### Author Response · Authors · 2023-11-21
> > > **Thanks for your careful review and helpful comments**
> > >
> > > Dear Reviewer Kntd,
> > >
> > > Thanks for your careful review and helpful comments, we have edited the explanation on updates to the revision in the [previous reply](https://openreview.net/forum?id=7ERQPyR2eb&noteId=90wH1FfhZX). Exactly, we have also corrected Figure 1, 2, 4 in the latest version, not only the mentioned Figure 3.
> > >
> > > Thanks for your professional suggestions and the helpful discussion!

---

### Official Review · Reviewer_MF4S · 2023-11-01

**Soundness:** 4 excellent
**Presentation:** 3 good
**Contribution:** 3 good
**Rating:** 10
**Confidence:** 4

**Summary:**

This paper proposes a novel method for synthesizing photorealistic talking portraits from a single source image coupled with either a speech signal or a driving facial video. The method generates a 3D avatar from the source image and then animates it based on the speech signal or the driving video. Apart from the talking head synthesis, the method pays attention in the realistic synthesis of torso movements and backgrounds. Regarding the module of 3D face reconstruction, the method adopts an image to plane model. Regarding the module of animation, the method introduces a novel facial motion adapter. The paper presents detailed experiments (qualitative and quantitative evaluations, user studies and ablation studies) that show the advantages and promising performance of the proposed method.

**Strengths:**

+ The proposed method is interesting and its pipeline has sufficient novelty, especially in terms of the combination of the large-scale image-to-plane backbone, the motion adapter and the Head-Torso-Background Super-Resolution model, which results in particularly realistic results.

+ The paper includes an in-depth experimental evaluation that provides sound evidence about the promising results of the proposed method. In more detail, the proposed method is compared with several recent SOTA methods (despite the fact that some additional methods should have been included - see comments below). The evaluation includes qualitative and quantitative comparisons, as well as user studies that are important to judge the perceived quality of the results. The ablation studies are also detailed and clearly show the importance and benefits of the different modules of the pipeline. Finally, the supplementary videos are informative and help to appreciate the visual quality of the results, as well as the advantages of the proposed method over the previous SOTA techniques.

**Weaknesses:**

- The paper has omitted citing some important related methods of the field:

J. S. Chung, A. Jamaludin, and A. Zisserman, “You said that?” in BMVC, 2017.

Ye, Z., Xia, M., Yi, R., Zhang, J., Lai, Y.-K., Huang, X., et al. (2022). Audio-driven talking face video generation with dynamic convolution kernels. IEEE Transactions on Multimedia.

In addition, the method is based on the projected normalized coordinate code (PNCC) representation but it has not cited one of the most important works of the field that is also based on the same representation:

Kim, H., Garrido, P., Tewari, A., Xu, W., Thies, J., Niessner, M., Pérez, P., Richardt, C., Zollhöfer, M. and Theobalt, C., 2018. Deep video portraits. ACM transactions on graphics (TOG), 37(4), pp.1-14.

Also, the method supports the conditioning of the animation based on a speech audio signal but it does not cite another important and seminal work of the field that does something similar:

Kim, H., Elgharib, M., Zollhöfer, M., Seidel, H.P., Beeler, T., Richardt, C. and Theobalt, C., 2019. Neural style-preserving visual dubbing. ACM Transactions on Graphics (TOG), 38(6), pp.1-13.


- In the part of the Audio-driven talking face generation of the experimental comparisons, some additional recent works should have been added in the comparisons. For example, the paper should have cited and included in the comparisons the following methods that solve the same problem:

Yi, R., Ye, Z., Zhang, J., Bao, H. and Liu, Y.J., 2020. Audio-driven talking face video generation with learning-based personalized head pose. arXiv preprint arXiv:2002.10137.

Yao, S., Zhong, R., Yan, Y., Zhai, G. and Yang, X., 2022. DFA-NeRF: Personalized talking head generation via disentangled face attributes neural rendering. arXiv preprint arXiv:2201.00791.

**Questions:**

- In the methodology, the paper does not clarify how the per-frame head pose is predicted from the audio, in the case of audio-driven talking face generation. More details and clarifications about that should be provided.

- In terms of the realism of the final synthetic video, one can observe that the method has some limitations regarding generating realistic background. Considering the general quality and realism of the results, this is an acceptable limitation, but the authors should openly discuss these limitations and link them with potential directions of future works.

**Details Of Ethics Concerns:**

The proposed method has the ability to generate realistic talking portrait videos. As all works that are related to deep fakes, this technology can be misused and generate a video of a person saying something that he/she has never said, without their consent.
The appendix of the paper includes a satisfactory discussion of the related ethical concerns and proposes mitigations that make sense.

---

> ### Author Response · Authors · 2023-11-16
> **Author Response to Reviewer MF4S (Part 1/2)**
>
> We are grateful for your positive review and valuable comments, and we hope our response fully resolves your concerns.
>
>
>
> > Q1: Should citing important related methods of audio-driven methods, other methods using PNCC, and some work that solve the same problem. Compare with AUD ("Audio-driven talking face video generation with learning-based personalized head pose") and DFA-NeRF.
>
>
> - A1: Thanks for your insightful comment, we have cited the mentioned related works in the revised manuscript, which we found has improved the soundness of the manuscript. We also compare with DFA-NeRF in [this link](https://real3dportrait.github.io/static/videos/Comparison_with_DFANeRF.mp4) and AUD in [this link](https://real3dportrait.github.io/static/videos/Comparison_with_DFANeRF.mp4). Note that both DFA-NeRF and AUD are person-specific methods that over-fit a target identity with a two-minute-long video. We tried our best to reproduce these results, but since DFA-NeRF seems incomplete, it could only produce reasonable results when driven by the ground truth expression. So the comparison is mainly performed with AUD. Firstly, we can perform a qualitative comparison from the demo videos. We can see that both DFA-NeRF and AUD produce temporal jittering results, and AUD has a slightly worse identity-preserving ability. By contrast, our method is one-shot, temporally stable, identity-preserving, and lip-synchronized. As for the quantitative comparison with AUD, for a fair comparison, we crop and downsample our method into 256x256 resolution to align with AUD. The results are as follows, we can observe better Sync Score, AED, CSIM, and FID over AUD.
>    - | Method | CSIM$\uparrow$   | FID$\downarrow$  | AED$\downarrow$ | Sync$\uparrow$   |
> | ------ | ---------------- | ---------------- | --------------------------- | ---------------- |
> | AUD    | 0.742            | 47.59            | 0.160                       | 5.863            |
> | Ours   | $\mathbf{0.775}$ | $\mathbf{42.63}$ | $\mathbf{0.141}$            | $\mathbf{6.630}$ |
>
>
>
>
> > Q2: In the methodology, the paper does not clarify how the per-frame head pose is predicted from the audio, in the case of audio-driven talking face generation. More details and clarifications about that should be provided.
> - A2: Thanks for pointing out this problem. In this paper, since the prediction of head pose is not our main interest, we devise a naive strategy to obtain head pose sequences from GT videos of CelebV-HQ. To be specific, (1) in the same-identity setting, the driving head pose is exactly the same as the GT video of the source image (source image is the first frame of the test video); (2) in the cross-identity setting, the driving head pose is extracted from the driving video; (3) then, in the audio-driven setting, most importantly, we first estimate the head pose in the source image, then we query 10 videos in the CelebV-HQ that has nearest distance between its first frame's head pose and the source image. Once the candidate videos that provide head pose are sampled, we randomly choose one of them to drive the source image. We acknowledge that an explanation of the head pose selection strategy is beneficial for the soundness of the manuscript, and have added the discussion above **in the blue-colored part in Appendix D.1 of the revised manuscript**. Thanks again for improving the soundness of this paper!

---

> > ### Author Response · Authors · 2023-11-16
> > **Author Response to Reviewer MF4S (Part 2/2)**
> >
> > > Q3: In terms of the realism of the final synthetic video, one can observe that the method has some limitations regarding generating realistic background. Considering the general quality and realism of the results, this is an acceptable limitation, but the authors should openly discuss these limitations and link them with potential directions of future works.
> >
> > - A3: We admit that the quality of the inpainted background is important for the realism of the final synthetic video. Since the background inpainting process can be performed by arbitrary off-the-shelf inpainting methods, we didn't focus on this topic in the original manuscript. We also notice that the proposed KNN-based inpainting method is relatively naive and can be replaced with a more advanced neural network-based inpainting method. We truly agree with your idea that an open discussion on the limitation of the background inpainting part is necessary and could motivate future improvements. Specifically, in the revised manuscript, we have added the following paragraph **to the blue-colored part in Appendix E.1, Limitation and future works, of the revised manuscript**:
> >
> >   > Though we have achieved generally high-quality realistic talking portrait results, one of the limitations is that the inpainted background image could leak unnaturalness when the talking person is doing a large pose motion. We believe the introduced naive KNN-based background inpainting method is to be blamed. Since generating a realistic background is of high importance to ensure the realism of the final video, we plan to upgrade the background inpainting method with a more advanced neural network-based system, such as MAT (MAT: Mask-Aware Transformer for Large Hole Image Inpainting) and DDNM (ero-shot image restoration using denoising diffusion null-space model). An alternative might be learning the background inpainting module within the HTB-SR model in an end-to-end manner. We leave this for future work.
> >
> >
> > ## Summary
> > Following your comments, we have modified the manuscript and we found the clarity and completion of the paper have been improved. Again, we thank the reviewer for the insightful review and "Accept" recommendation for our paper.

---

> ### Author Response · Authors · 2023-11-22
> **Hoping that our response could address your concern**
>
> Dear Reviewer MF4S,
>
> Thank you again for your time and effort in reviewing our work! We would appreciate it if you can let us know if our response has addressed your concern. As the end of the rebuttal phase is approaching, we look forward to hearing from you and remain at your disposal for any further clarification that you might require.

---

> ### Author Response · Authors · 2023-11-23
> **Dear Reviewer,**
>
> Dear Reviewer MF4S.
>
> As the discussion period is closing in several hours, we would like to know if there are any additional questions. We are glad to answer them.
>
> Again, we sincerely appreciate your insightful review and "Accept" recommendation for our paper.

---

### Official Review · Reviewer_8dWt · 2023-11-04

**Soundness:** 2 fair
**Presentation:** 3 good
**Contribution:** 1 poor
**Rating:** 8
**Confidence:** 3

**Summary:**

This paper proposes a method to create a reanimatable avatar from a single image. Leveraging the latent space of EG3D, the method learns to generate a canonical tri-plane from a given input image. This canonical triplane is then deformed using motion module conditioned on projected normalized co-ordinate code (PNCC). Finally, the deformed triplane is volume rendered to generate the final image. The motion conditioned tri-plane representation generates good results and outperforms some prior art, but the overall architecture and design is very similar to HiDe-NeRF.

**Strengths:**

1) The paper is well written.

2) The qualitative results on reanimation are good, even when driven by audio.

3) The quantitative results show that the proposed method out-performs prior art.

4) The background is rendered well and merges seamlessly with the foreground.

**Weaknesses:**

1) Given that the overall architecture is very similar to HiDe-NeRF, it is unclear where the improvement of the proposed method is coming from. Is it because a pretrained Tri-plane is a better representation than the multi-resolution tri-plane features of HiDe-NeRF? It would be great if the authors could clarify this

2) The addiction of background and torso modelling, while important, is relatively incremental.

**Questions:**

It would be great if the authors could clarify why the proposed method works better than HiDe-NeRF despite having a very similar architecture and set-up. Is pretraining the only reason?

---

> ### Author Response · Authors · 2023-11-16
> **Author Response to Reviewer 8dWt (Part 1/2)**
>
> We thank the reviewer for the constructive feedback and the positive remarks on our qualitative/quantitative results. We acknowledge that your concerns are mainly about the novelty of this paper, and hope our response resolves your concerns fully.
>
> > Q1: (1) Given that the overall architecture is very similar to HiDe-NeRF, it is unclear where the improvement of the proposed method is coming from. (2) Is it because a pretrained Tri-plane is a better representation than the multi-resolution tri-plane features of HiDe-NeRF? It would be great if the authors could clarify this
>
> - A1: (1) Actually, there are several significant differences between our method and HiDe-NeRF. Firstly, we pretrain the I2P model while HiDe-NeRF doesn't; secondly, we use the motion adapter to predict a motion diff-plane while HiDe-NeRF resorts to the deformation field; thirdly, we first propose to individually model the head-torso-background parts; fourthly, we propose a generic audio-to-motion model to additionally support audio-driven tasks while HiDe-NeRF only support video-driven.
> - (2) As for the performance improvement of ours over the HiDe-NeRF, we think the reasons are three-fold. Firstly, as pointed out by the reviewer, pre-training the I2P model is an important reason since it utilizes the prior knowledge about 3D geometry from the 3D GAN. We have ablated this in Figure 15 of the original manuscript. **Secondly, and most importantly, we think replacing the deformation field with our proposed motion adapter is the key reason.** The deformation field aims to learn to deform the ray in the canonical space during inference to achieve dynamic scene synthesis. However, based on our experiment (please kindly refer to [this link](https://real3dportrait.github.io/static/videos/Comparison_with_deformation.mp4), in which we compare our motion adapter versus the deformation field by HiDe-NeRF), we find deformation field destroys the good 3D mesh reconstructed by our I2P model and tends to predict an ill-posed face surface (instead of a 3D mesh), which results in bad robustness and bad visual quality. By contrast, our motion adapter only edits the minimal part of the 3D face representation corresponding to the input PNCC, hence maintaining the most of geometry and texture information stored in the reconstructed canonical tri-plane. As a result, our method achieves better identity preserving and geometry realness than HiDe-NeRF. The third reason for our better performance over HiDe-NeRF is that we individually model the torso part. By contrast, in HiDe-NeRF the torso part will rotate along with the head part, which is unrealistic. We sincerely thank the reviewer for sharing the valuable feedback, and we have added the discussion above in the appendix of the revised manuscript (**Please refer to the blue-colored part in Appendix A of the revised manuscript for more details**), which we suspect could better clarify our novelty and the intrinsic mechanism of our better quality than HiDe-NeRF.

---

> ### Author Response · Authors · 2023-11-16
> **Author Response to Reviewer 8dWt (Part 2/2)**
>
> > Q2: The addition of background and torso modelling, while important, is relatively incremental.
>
> A2: We appreciate it for acknowledging the importance of individually modeling the torso and background segment in the field of one-shot 3D talking generation. We believe that the proposed head-torso-background individual modeling paradigm could pave the way for future works to achieve more realistic results (since most existing works only focus on the head part or jointly model the head and torso). As for the "incremental", we admit that our HTB-SR has adopted many known techniques to guarantee robustness and performance, e.g., the Torso branch is based on the warping-based method. However, we have also come up with several necessary modifications to achieve more realistic results. For instance, in the warping-based torso branch, we propose to replace the implicit key points in the warping-based method with our predefined projected facial landmark, which is necessary to guarantee temporal stability. Also, it is the first attempt that only warps a specific torso segment of the talking person frame, as the previous methods typically warp the whole frame, which results in distortion artifacts in the background part. Besides, when integrating the head-torso-background segments, we propose an alpha-blending-style fusion mechanism, which could handle the robustness issue caused by the naive channel-wise concatenation. We hope the above clarification could better highlight our contribution to the addition of background and torso modeling.
>
> ## Summary
> In summary, following the given comments, we have performed in-depth analysis and experiment to clarify the intrinsic mechanism of our method's advantage over HiDe-NeRF. We also rethink the novelty of the addition of background and torso modelling. With these revision, we find that the soundness and readability of the paper has been improved. Again, we would like to appreciate the reviewer’s valuable review. We sincerely hope the reviewer will reconsider their rating in light of the rebuttal.

---

> > ### Comment · Reviewer_8dWt · 2023-11-21
> > **Rebuttal Response**
> >
> > I would like to thank the authors for their response and clarification regarding the contributions. The motion adapter does seem to improve a depth a little bit, but a single example is not enough. It’d be great if a few more examples could be provided along with a visualization of the normals, since it is hard judge the geometry using only depth.

---

> ### Author Response · Authors · 2023-11-21
> **Mesh Visualization Updated**
>
> Dear Reviewer 8dWt,
>
> Thanks for your acknowledgment on our clarification regarding the contributions. As for the remaining problem "visualization of the normals". We admit that depth alone is not adequate to prove the geometry quality, so a mesh visualization, which could reflect both depth and surface normals information is necessary.
>
> To this end, we have updated a demo video [in this link](https://real3dportrait.github.io/static/videos/Visualiztion_MRC_Mesh.mp4) on our demo page, which compares the visualized mesh from Real3D-Portrait and Deformation Field. The visualized mesh contains the geometry information of depth and surface normals. We can see that our method with I2P model and Motion Adapter could generate mesh with good geometry while the Deformation cannot.
>
> As for the details in our visualization proces, we mainly follow the instruction of EG3D [in this link](https://github.com/NVlabs/eg3d#generating-media). Specifically, we first sample the density in a $[-1,1]^3$ cube of the world coordinate, then export a .mrc file, which is then visualized by a software named ChimeraX.
>
> Finally, we'd like to thank you for your precious time and valuable comments, which have improved the soundness and clarity of this manuscript. We will be very happy to clarify any remaining points (if any).

---

> > ### Comment · Reviewer_8dWt · 2023-11-21
> > **Visualization Response**
> >
> > I would like to thank the authors for responding promptly with these visualizations. I would strongly encourage them to include these visualization for more subjects in the final version of the paper. I have updated my score.

---

> > > ### Author Response · Authors · 2023-11-21
> > > **Thanks for your acknowledgement on our rebuttal**
> > >
> > > Dear reviewer 8dWt,
> > >
> > > Thanks for your acknowledgment of our rebuttal. We'd like to add more visualization results of more subjects in the camera-ready.
> > >
> > > Finally, we'd like to thank you for the professional suggestions and deep insights that improved the soundness and readability of the paper.

---

### Official Review · Reviewer_yTCz · 2023-11-10

**Soundness:** 3 good
**Presentation:** 4 excellent
**Contribution:** 2 fair
**Rating:** 8
**Confidence:** 5

**Summary:**

This work describes a method for single-shot 3D reconstruction and animation of a given 2D facial image. It uses an image to plane method to lift a 2D face into the tri-plane of a 3D GAN. It then morphs the triplane by adding a delta tri-plane which encoded the target expression via an PNCC encode image to animate the face. Differently from prior work the proposed method provides a solution for both video and audio driven animation of the face and also provides a super-resolution module for correctly morphing the torso and in-filling the exposed background regions. The proposed method is compared to several existing baseline video and audio-driven methods and shown to be superior them.

**Strengths:**

The paper addresses a fairly novel problem of single-shot joint 3D reconstruction and animation of facial images. Differently from prior work it seeks to inject the animation information directly into the 3D representation versus the dominant approach of animating in 2D first and then lifting into 3D. As the authors correctly point out animation in 3D versus 2D results is more correct handling of large head poses and less warping artifacts. So this is an important problem to address towards enabling large head pose facial talking head generation.

In comparison to the existing works on the topic, this work introduces a joint framework for both video and audio driven animation of 3D facial representations by encoding the audio signal into a PNCC representation. It also does a nice job of proposing to handle the torso and the background as a part of the overall solution thus enabling a more production-ready complete end-to-end framework. This is an often neglected detail in many works that treat the head in isolation from the backgrounds in which it exists requiring additional pre and post processing steps to deal with the torso and background.

**Weaknesses:**

1. The method by design requires the canonicalization, i.e., removal of the source image's facial expression, for the driving expression to be successfully applied to it. This is because the PNCC code is derived purely from the target driving video/audio's expression and hence cannot contain the information to erase/neutralize the source image's facial expression. I think the proposed method achieves this canonicalization during the fine-tuning phase with the Celeb-V-HQ video dataset. However, in my experience without any further stronger constraints, the proposed method cannot fully remove the source image's original facial expression. Can the author show examples of these canonicalized 3D face reconstructions of the input images without applying the target expression? Related to this is the question of how robust is the proposed method to the presence of large facial expressions in the source image. Does it work for expressions where the source image has a wide open mouth and lowered jaw or nearly closed eyes, for example?

2. Overall the proposed methods is an obvious combination of several existing ideas from prior works on the topic to culminate in a successful large engineered end-to-end solution. For example, the idea of using I2P was previously proposed in LP3D; the idea of using rendered target expression images from a 3DMM was proposed in Li et al., 2023b, the audio to PNCC code predictor is borrowed from prior work; and the torso warping module is also borrowed from Wang et al., 2021. While the proposed combination of existing ideas results in an effective solution, from the research perspective the overall solution is light on significant novel or surprising insights.

3. The authors don't specify the dataset/protocol that they used for evaluation.

4. Comparisons of the proposed method to several of the newer (than Face-Vid2Vid) and better performing video-driven 2D facial animation methods are missing. Below I list several of them.

[1] Thin-plate spline motion model for image animation., CVPR 2022

[2] Depth-Aware Generative Adversarial Network for Talking Head Video Generation., CVPR 2022

[3] FNeVR: Neural volume rendering for face animation., NeurIPS 2022

[4] Latent Image Animator: Learning to Animate Images via Latent Space Navigation., ICLR 2022

[5] DPE: Disentanglement of Pose and Expression for General Video Portrait Editing., CVPR 2023

[6] Conditioned Memory Compensation Network for Talking Head video Generation., ICCV 2023

**Questions:**

I would like to see the authors' response to the questions I have raised in the weaknesses section of the paper above.

---

> ### Author Response · Authors · 2023-11-16
> **Author Response to Reviewer yTCz (Part 1/2)**
>
> We are grateful for your positive review and valuable comments, and we hope our response fully resolves your concerns.
>
> > Q1: (1) Doubt about the motion adapter's capability of expression canonicalization (or say, neutralization), as without any further stronger constraints, the proposed method cannot fully remove the source image's original facial expression. (2) Can the author show examples of these canonicalized 3D face reconstructions of the input images without applying the target expression? (3) Does it work for expressions where the source image has a wide open mouth and lowered jaw or nearly closed eyes, for example?
>
> - A1: (1) Thanks for pointing out this problem! We acknowledge your concerns that if the motion adapter is only provided with the target PNCC, it cannot work well with the I2P model which could have arbitrary expression in the cano_planes. We apologize for not clarifying this issue in the previous manuscript. Actually, our key motivation in inventing the motion adapter is to directly perform the source-to-target transform in the cano_planes (skipping an intermediate expression canonicalization process that is required by a traditional source-to-neutral-to-target pipeline). In practice, instead of only feeding the target PNCC into the motion adapter, the input is a channel-wise concatenated source-target PNCC image of shape $(C_{src}+C_{tgt})\times H\times W$. So the motion adapter could learn the source-to-target transform only related to the 3DMM geometry prior while eliminating any other texture-related information. We sincerely thank you for your insightful comments and we have added the technical details above to the revised manuscript (**Please refer to the blue-colored part in Section 3.2 of the revised manuscript for more details**).
> - (2) Sure, in [this link](https://real3dportrait.github.io/static/videos/I2P_multi_view_synthesis.mp4) we provide a demo in which the I2P model is extracted from Real3D-Portrait's final checkpoint and perform the multi-view synthesis task. We can observe that the rendered canonical plane indeed is of the source expression. (3) Yes, as we discussed in the first point, although without an extra expression neutralization process since our motion adapter learns the source-to-target transform by utilizing the input source-target-concatenated PNCC, it could support arbitrary target expression given a source image of arbitrary source expression. To demonstrate this, we provide a demo video in [this link](https://real3dportrait.github.io/static/videos/Demo_critical_source_images.mp4), in which we show our method performs well given the critical source images that have a wide open mouth, lowered jaw, or totally closed eyes. To summarize, in this QA, we discussed Real3D-Portrait's generalizability of 3D reconstruction and animation. We'd like to thank you for providing constructive comments to help us enrich the technical details in the manuscript and motivate us to produce these intuitive demos, which we believe are helpful for new readers to perceive the intrinsic mechanism of this paper.
>
>
>
> > Q2: About novelty. The proposed methods combines several existing ideas from prior works to culminate in a successful large engineered end-to-end solution.
>
> - A2: Thanks for your comments. We'd like to explain the major novelty of this paper as follows. We first propose an efficient motion adapter that achieves high-quality and geometry-preserving 3D face source-to-target animation, without an extra expression neutralization process. Secondly, we are the first work that points out the necessity of head-torso-background individual modeling and come up with an effective HTB-SR model (with several necessary modifications, such as explicit keypoint in the torso branch and alpha-blending-style fusion mechanism in combining the three segments) to handle this issue. Thirdly, we are the first work that achieves audio-driven ont-shot 3D talking face. Besides, the proposed four-stage framework, which supports audio/video-driven applications,  is easy to be expanded to support other sub-topics, such as stylized talking faces, etc. We expect Real3D-Portrait to provide a new paradigm for the new generation of one-shot 3D talking face methods to produce realistic video results.

---

> ### Author Response · Authors · 2023-11-16
> **Author Response to Reviewer yTCz (Part 2/2)**
>
> > Q3: Specify the dataset/protocol that are used for evaluation.
>
> - A3: Thanks for pointing out this issue. Due to space limitations, in the original manuscript, we only briefly explain the dataset/protocol used for evaluation. To be specific, there are three groups of evaluated data: (1) for the Same-Identity Reenactment, we randomly chose 100 videos in our preserved validation split of CelebV-HQ. (2) for the Cross-Identity Reenactment, we use the first frame of the previously selected videos to obtain 100 identities, then use the expression-pose sequence from a randomly selected video to construct the cross-identity reenactment data pair. (3) for the audio-driven scenario, we use 10 out-of-domain images downloaded from the internet (which are exactly the 10 identities shown in the demo video) and choose 10 audios from different languages to form the data pair (so each method has 100 videos as the test samples).  We apologize for the missed dataset/protocol specification in the original manuscript, and **we have added it to the blue-colored part in Appendix D.1 of the revised version**. Thanks for your suggestion that improves the technical soundness of the manuscript.
>
>
> > Q4: Add comparison to newer (than Face-Vid2Vid) and better performing video-driven 2D facial animation methods.
>
> - A4: We admit that more recent 2D methods that are newer Face-Vid2Vid should be compared for better soundness. We have tested 5 of the 6 mentioned baselines (We failed to reproduce FNeVR since the checkpoint link provided in the official repo is broken). We provide a qualitative comparison demo video in [this link](https://real3dportrait.github.io/static/videos/Comparison_with_5_additional_VD_baselines.mp4), in which we can observe that (1) both the advanced 2D methods and our real3d-portrait achieve realistic results when the head motion is small (refer to the part of Obama in the demo video); (2) most of the tested 2D methods failed to achieve naturalness when given a large head pose motion (refer to the part of KFC in the demo) while our Real3D-Portrait keeps performing well. As for more quantitative results, due to the time limitation, we have come up with the quantitative result of "Thin-plate spline" (TPS) and DPE, which are widely used 2D methods that are considered the successor of Face-Vid2Vid. The tested results are as follows, we can see that our Real3D-Portrait outperforms TPS in terms of. We plan to add the quantitative results of the other 4 additional baselines in the camera-ready version.
>
>   - | Method | CSIM$\uparrow$   | FID$\downarrow$  | AED$\downarrow$  | APD$\downarrow$  |
>   | ------ | ---------------- | ---------------- | ---------------- | ---------------- |
>   | TPS    | 0.745            | 43.54            | 0.137            | 0.028            |
>   | DPE    | 0.731            | 44.07            | 0.135            | 0.021            |
>   | Ours   | $\mathbf{0.764}$ | $\mathbf{41.58}$ | $\mathbf{0.129}$ | $\mathbf{0.017}$ |
>  Note that in the above table, we resample our method from 512x512 to 256x256 resolution for fair comparison with the baselines since they produce 256x256 results.
>
> ## Summary
> Following your comments, we have modified the manuscript and we found the soundness and completion of the paper have been improved and the novelty has been more highlighted. Again, we thank the reviewer for the insightful review and positive recommendation for our paper.

---

> > ### Comment · Reviewer_yTCz · 2023-11-22
> > **Quantitative comparison results to MCNet**
> >
> > Can the authors additionally provide quantitative comparison results of your method versus MCNet (ICCV 2023)?

---

> ### Author Response · Authors · 2023-11-22
> **Hoping that our response could address your concern**
>
> Dear Reviewer yTCz,
>
> Thank you again for your time and effort in reviewing our work! We would appreciate it if you can let us know if our response has addressed your concern. As the end of the rebuttal phase is approaching, we look forward to hearing from you and remain at your disposal for any further clarification that you might require.

---

> ### Comment · Reviewer_yTCz · 2023-11-22
> **Response to authors**
>
> For the lowered jaw expression in the source image, I meant an expression like this: https://www.shutterstock.com/image-photo/close-portrait-happy-young-girl-dressed-772513501. The results that you've provided are for an "face look down" expression. Can the authors provide results for a source image like the one shared?

---

> > ### Author Response · Authors · 2023-11-22
> > **Demo for lowered yaw expression**
> >
> > Dear Reviewer yTCz,
> >
> > Thanks for providing a source image of the lowered yaw expression, and we apologize for the misunderstanding of the definition of the lowered yaw expression. We have updated the demo video of the given source image, please kindly refer to the demo [in this link](https://real3dportrait.github.io/static/videos/Demo_lowered_yaw_expression.mp4). We can see that our I2P model and Motion Adapter generalize well to the out-of-domain source image with a challenging lowered yaw expression.
> >
> > As for the quantitative comparison results of our method versus MCNet (ICCV 2023), please stay tuned and we will come up with the results as soon as possible.
> >
> > Again, we'd like to thanks for your precious time and helpful suggestions!

---

> > > ### Comment · Reviewer_yTCz · 2023-11-22
> > > **Thanks!**
> > >
> > > Nice! the video results for the portrait with the "lowered jaw" look great.
> > >
> > > I look forward to your comparison results to MCNet.

---

> ### Author Response · Authors · 2023-11-22
> **Updated quantitative comparison with MCNet**
>
> Dear Reviewer yTCz,
>
> We acknowledge that it is necessary to quantitatively compare our 3D-based method with the most recent 2D-based SOTA, MCNet (ICCV 2023). We have updated the quantitative results of MCNet (ICCV 2023) as follows. We can see that MCNet achieves better CSIM and FID than the other 2D baselines, but is slightly inferior to our method. We suspect the hard large-pose cases in the test samples are the reason, as the APD (average pose distance) is relatively worse.
>
> | Method | CSIM$\uparrow$   | FID$\downarrow$  | AED$\downarrow$  | APD$\downarrow$  |
> | ------ | ---------------- | ---------------- | ---------------- | ---------------- |
> | TPS    | 0.745            | 43.54            | 0.137            | 0.028            |
> | DPE    | 0.731            | 44.07            | 0.135            | 0.021            |
> | MCNet  | 0.758            | 42.49            | 0.136            | 0.030            |
> | Ours   | $\mathbf{0.764}$ | $\mathbf{41.58}$ | $\mathbf{0.129}$ | $\mathbf{0.017}$ |
>
> We find the comparison with these state-of-the-art 2D methods helpful to demonstrate the performance of our method. We have added the comparison table in **Appendix D.5** of the latest manuscript, and have cited the mentioned remarkable works in our discussion on related work (The **paragraph named "2D/3D Face Animation" of Sec. 2** in the revised manuscript).
>
> Finally, we'd like to thank you for your expertise and valuable comments, with which we found have improved the completeness of the related work part and the soundness of the experiment part by a large margin.

---

> > ### Comment · Reviewer_yTCz · 2023-11-23
> > **Thanks!**
> >
> > Acknowledging receipt of these results. Thank you. They look great.

---

> > > ### Author Response · Authors · 2023-11-23
> > > **Thanks for your acknowledgment of our rebuttal**
> > >
> > > Dear Reviewer yTCz,
> > >
> > > Thanks for your acknowledgment of our rebuttal!
> > >
> > > Again, we'd like to thank you for your expertise and engaged discussion that improved the soundness and completeness of the paper.

---

### Author Response · Authors · 2023-11-16
**General Response to All Reviewers**

We would like to thank the reviewers for their constructive reviews! Following the comments and suggestions of reviews, we have revised the manuscript, and the revised parts are marked in blue. We have also uploaded 7 new demo videos on the [demo page](https://real3dportrait.github.io/). Here we summarize the revision as follows:

- We provide two demo videos to demonstrate that our I2P could reconstruct the 3D face mesh ([Rebuttal Demo 2](https://real3dportrait.github.io/static/videos/Comparison_with_deformation.mp4) on the demo page) and the motion adapter could effectively transform the 3D face into the target expression ([Rebuttal Demo 4](https://real3dportrait.github.io/static/videos/I2P_multi_view_synthesis.mp4)) on the demo page, which is suggested by Reviewer 8dWt, Kntd, and yTCz.
- To further show the generalizability of our I2P+MA to hard source images, as suggested by Reviewer yTCz, we provide [Rebuttal Demo 1](https://real3dportrait.github.io/static/videos/Demo_critical_source_images.mp4) on the demo page to show that our method generalizes well to critical cases such as large-opened mouths and closed eyes.
- As suggested by Reviewer MF4S, Kntd, and yTCz, we additionally compare 6 video-driven baselines  (comparison with five 2D methods in [Rebuttal Demo 5](https://real3dportrait.github.io/static/videos/Comparison_with_5_additional_VD_baselines.mp4), comparison with one 3D method in [Rebuttal Demo 3](https://real3dportrait.github.io/static/videos/Comparison_with_Next3D.mp4) on the demo page) and 2 audio-driven baselines (comparison with one 2D method in [Rebuttal Demo 6](https://real3dportrait.github.io/static/videos/Comparison_with_AUD.mp4), comparison with one 3D method in [Rebuttal Demo 7](https://real3dportrait.github.io/static/videos/Comparison_with_DFANeRF.mp4) on the demo page).
- In Section 3.2, we clarify the input of the Motion adapter, as suggested by Reviewer Kntd and yTCz.
- In Section 3.4, we correct the statement about 3DMM expression bases, and change the reconstruction loss of VAE from Landmark MSE to Expression MSE, as suggested by Reviewer yTCz.
- In Appendix A, we additionally discuss the comparison between our method and the two most related baselines, HiDe-NeRF and GOS-Avatar, in detail, as suggested by Reviewer 8dWt and Kntd. By clarifying the difference, we make it more clear for the new readers to clarify the novelty of this paper.
- In Appendix D.1, we add a section to provide details about the evaluation process and the selecting strategy of camera pose, as suggested by Reviewer yTCz and MF4S.
- In Appendix E.1, we add a discussion on the limitation of the background inpainting process, as suggested by Reviewer MF4S.
- Additionally cited several important prior works in the field of audio-driven talking face generation, as suggested by Reviewer MF4S.

Thanks again for the reviewers' great efforts and valuable comments, which have improved the soundness of the manuscript. We have carefully addressed the main concerns and provided detailed responses to each reviewer. We hope you will find the responses satisfactory. We would be grateful if we could hear your feedback regarding our answers to the reviews.

---

### Author Response · Authors · 2023-11-20
**Dear AC and Reviewers,**

Thanks again for your great efforts and valuable comments.

We have carefully addressed the main concerns and provided detailed responses to each reviewer. We hope you might find the responses satisfactory. As the end of the rebuttal phase is approaching, we would be grateful if we could hear your feedback regarding our answers to the reviews. We will be very happy to clarify any remaining points (if any).

---

### Comment · Reviewer_MF4S · 2023-12-04
**My final rating**

After reading the other Reviewers' comments as well as the detailed responses by the Authors, I am upgrading my final recommendation. I was completely satisfied by the Authors' responses, since they have successfully addressed all concerns by all the Reviewers, with clarifications, meaningful amendments and detailed additional experiments. It is now clear to me that this is a high-quality paper.

---

### Meta-Review · Area_Chair_hgSQ · 2023-12-06

**Metareview:**

This paper describes a method for generating 3D reconstructions and talking face animations from a given 2D facial image. In the first step, the method utilizes an Image-to-Plane (I2P) model to canonicalize 3D faces in tri-plane representation. By encoding the audio signal into a PNCC representation, the paper presents a joint framework for both video and audio-driven animation of 3D facial representations. In experiments, the proposed method outperforms several video and audio-driven methods.

Among the strengths, the reviews note that the single-shot joint 3D reconstruction and talking head animation from a single facial image is novel. The novelty of this approach lies in injecting the animation information directly into the 3D representation rather than animating in 2D first and lifting it into 3D like most previous approaches. Both qualitatively and quantitatively, the experiment shows promising results and outperforms compared methods.

Several concerns were raised in the initial reviews. (1) The method requires canonicalization to fully remove the original facial expression. There is doubt in the review as to whether it is possible and there is a request for results of specific expressions. (2) The proposed method combines several existing ideas. In particular, a review mentioned that the architecture is similar to HiDe-NeRF. Therefore, the novelty appears to be incremental. (3) There is no comparison with several newer methods in the paper. (4) The paper omits some important references. (5) Some design decisions may not be well-supported. As a result of the rebuttal, further clarification is provided regarding the contributions, differences with HiDe-NeRF, the use of motion adapters for canonicalization, more comparisons with the methods requested by reviews, examples of the expression requested, and demonstrating the differences between the deformation field and residual motion diff-plane. The rebuttal successfully addressed the majority of concerns. The only remaining reservation by a review is that, although cleverly using several well-known ideas for designing an effective system with new functionality, the paper does not provide much surprising or novel insight.

**Justification For Why Not Higher Score:**

According to a review, although the paper cleverly uses several well-known concepts for designing an effective system, it offers little in the way of new insights or surprises. While the differences with previous methods are sufficient to qualify as novel, the paper does not possess the level of novelty or significance required for ICLR oral presentations.

**Justification For Why Not Lower Score:**

The paper is the first to propose a joint framework for generating 3D reconstructions and talking face animations from a given 2D facial image. Creating 3D talking heads has recently received wide attention, and the paper enables new applications. Technically, the paper presents some novel ideas, such as replacing the deformation field with the proposed effective motion adapter, modeling the head-torso-background parts, and proposing a generic audio-to-motion model. Both qualitatively and quantitatively, the proposed method outperforms a set of compared methods. There is strong support for the paper from all reviewers, as all of them give it a grade higher than 8 (accept, good paper). The revised paper has been generally regarded as a high-quality publication by all of them.

---

### Decision · Program_Chairs · 2024-01-16

Accept (spotlight)